# Cancer Survival Analysis via Zero-shot Tumor Microenvironment Segmentation on Low-resolution Whole Slide Pathology Images

**Jiao Tang, Wei Shao,**∗ **Daoqiang Zhang**
The College of Artificial Intelligence, Nanjing University of Aeronautics and Astronautics
The Key Laboratory of Brain-Machine Intelligence Technology, Ministry of Education
`tangjiao@nuaa.edu.cn, shaowei20022005@nuaa.edu.cn, dqzhang@nuaa.edu.cn`

## Abstract

The whole-slide pathology images (WSIs) are widely recognized as the golden standard for cancer survival analysis. However, due to the high-resolution of WSIs, the existing studies require dividing WSIs into patches and identify key components before building the survival prediction system, which is time-consuming and cannot reflect the overall spatial organization of WSIs. Inspired by the fact that the spatial interactions among different tumor microenvironment (TME) components in WSIs are associated with the cancer prognosis, some studies attempt to capture the complex interactions among different TME components to improve survival predictions. However, they require extra efforts for building the TME segmentation model, which involves substantial annotation workloads on different TME components and is independent to the construction of the survival prediction model. To address the above issues, we propose ZTSurv, a novel end-to-end cancer survival analysis framework via efficient zero-shot TME segmentation on low-resolution WSIs. Specifically, by leveraging tumor infiltrating lymphocyte (TIL) maps on the 50x down-sampled WSIs, ZTSurv enables zero-shot segmentation on other two important TME components (*i.e.,* tumor and stroma) that can reduce the annotation efforts from the pathologists. Then, based on the visual and semantic information extracted from different TME components, we construct a heterogeneous graph to capture their spatial intersections for clinical outcome prediction. We validate ZTSurv across four cancer cohorts derived from The Cancer Genome Atlas (TCGA), and the experimental results indicate that our method can not only achieve superior prediction results but also significantly reduce the computational costs in comparison with the state-of-the-art methods.

## 1 Introduction

Histopathology image analysis is a vital technology for cancer survival analysis [1, 2]. Traditional methods of survival analysis rely heavily on manual interpretation of these images by pathologists, which is time-consuming and prone to inter-observer variability. To address these challenges, computational approaches have been explored to assist the analysis process. Early computer-aided methods focus on handcrafted features extracted from specific regions of interests (ROIs), which are limited in scalability and generalization ability [3]. Recently, with the rapid development of the deep learning technology, training deep learning based whole-slide pathology image (WSI) analysis models for cancer survival prediction has gained significant attentions [4, 5]. However, the main challenge for survival analysis from the WSIs is that a high-resolution WSI is with large size (*e.g.,*

---

∗Corresponding author

39th Conference on Neural Information Processing Systems (NeurIPS 2025).

100,000-by-100,000 pixels), and thus it is impractical to directly feed them into deep neural networks due to memory limitations.

To make the analysis of WSIs memory-efficient, most of the existing studies firstly divide WSIs into multiple patches and identify key components before constructing the survival prediction model [6, 7, 8, 9, 10, 11]. Then, the patch-level representations are aggregated using attention [12] or pooling [13] strategies to predict the clinical outcome. However, dividing high-resolution WSIs into patches is computationally expensive, and the identified key components are insufficient to reflect the heterogeneous tumor microenvironment (TME) components and their spatial associations. As a highly heterogeneous disease, the progression of tumor is not only achieved by unlimited growth of the tumor cells, but also supported, stimulated, and nurtured by the TME components around (*i.e.,* stroma and lymphocyte) [14, 15].

For the purpose of capturing the spatial organizations among different TME components, the existing studies usually adopt graph neural networks (GNNs) to model the interactions among different TME components [16, 17, 18, 19, 20, 21, 22]. However, these methods require building the TME segmentation model for distinguishing different TME components before graph building, which need extra annotation efforts from the pathologists [3, 2, 23, 24]. Moreover, the existing studies train the TME segmentation and survival prediction models independently, which ignores the fact that the survival information could provide additional information to guide the TME segmentation task. For instance, we have more chance to see the TME component of lymphocyte near the tumor region for long survival patients, since it is widely recognized that the brisk interactions between lymphocyte and tumor regions will indicate a better clinical outcome [25]. Additionally, the existing GNN-based studies only extract visual features of each TME components as node representations while overlooking their corresponding semantic information, which limits the model's ability to distinguish different TME components for graph learning.

Based on the above considerations, in this paper, we propose ZTSurv, a novel end-to-end framework for cancer survival analysis through zero-shot segmentation of TME components on low-resolution WSIs. Specifically, instead of working on the high-resolution WSIs, ZTSurv leverages tumor infiltrating lymphocyte (TIL) maps on the 50x down-sampled WSIs to perform pixel-level zero-shot segmentation on other two important TME components ( *i.e.,* tumor and stroma) through pathology-language foundation model (*i.e.,* PLIP). Then, based on the visual and semantic information extracted from different TME components, we construct a heterogeneous graph to capture their spatial intersections for clinical outcome prediction. Extensive experiments on four cancer cohorts derived from The Cancer Genome Atlas (TCGA) validate the effectiveness of ZTSurv, revealing its superior predictive performance and computational efficiency in comparison with the state-of-the-art approaches.

We summarize our main contributions as follows:

1. We propose a novel end-to-end framework ZTSurv for survival prediction of human cancers that can simultaneously segment different TME components and capture their spatial interactions for clinical outcome prediction.

2. We develop a zero-shot TME segmentation method that can leverage TIL maps to segment other two TME components (*i.e.,* tumor and stroma) by the aid of pathology-language foundation model, which can reduce the pixel-level annotation efforts on different types of TME components for constructing the semantic segmentation model.

3. Instead of working on the high-resolution WSIs, we implement our ZTSurv on the 50× down-sampled WSIs that can significantly reduce the computational cost in comparison with the existing studies.

4. We incorporate semantic information alongside the visual features to represent each TME component for graph construction, which can more effectively distinguish different TME components.

## 2 Related Work

### 2.1 Survival Analysis in Gigapixel WSIs

Gigapixel WSIs provide crucial insights for cancer prognosis but are challenging to process directly due to their large size [26]. Existing studies typically divide WSIs into multiple fixed-size patches and

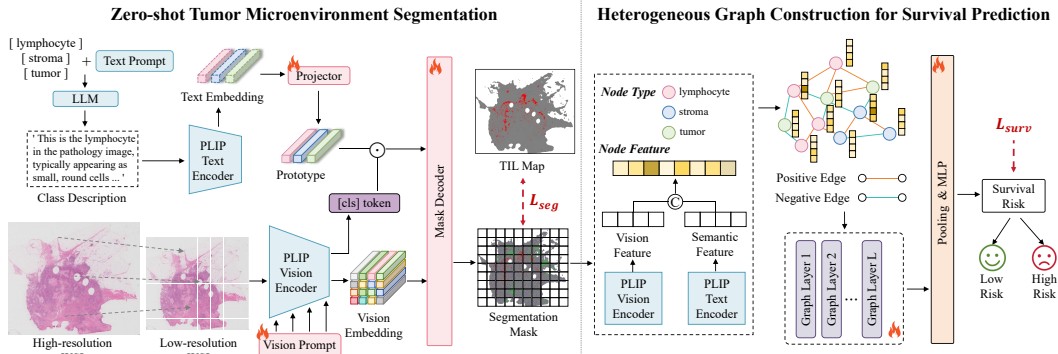

Figure 1: The overview of our proposed end-to-end framework ZTSurv for survival prediction of human cancers, which can simultaneously segment key TME components and capture their interactions for clinical outcome prediction.

extract useful information from these patches for survival prediction [6, 7, 27, 28, 8, 9, 11, 12, 13]. For instance, Mobadersany *et al.* [27] have presented a CNN-based survival prediction model based on the annotated ROIs extracted from WSIs. Zhu *et al.* [6] exploited and utilized all discriminative patches in WSIs to predict patients' survival status. Based on [6], Yao *et al.* [12] further employed an attention-based aggregation strategy to fuse informative patches for survival prediction. However, these methods fail to capture long-range spatial interactions of different patches in WSIs. To address the above challenges, a bunch of graph learning based survival prediction models are presented. For instance, Li *et al.* [21] proposed to model WSIs as graphs and then developed a graph convolutional neural network (graph CNN) with attention learning that better served the survival prediction by rendering the optimal graph representations of WSIs. Chen *et al.* [16] presented a spatially-resolved GCN [29] which hierarchically aggregated patch-level features to model local and global topological structures in WSIs. Di *et al.* [19] proposed a big-hypergraph factorization neural network to encode the correlation among vertices and hyperedges into two low-dimensional latent semantic spaces for better survival analysis. However, most of the graph learning studies overlook to discuss the interactions among different TME components that are important for cancer prognosis [25]. Also, they usually work on the high-resolution WSIs that will bring significant computational burden.

## 2.2 Capturing TME Heterogeneity for Survival Analysis

Recent studies have shown that capturing the heterogeneity of the TME is critical for improving the accuracy of survival prediction, as different TME components and their spatial interaction play important roles for patient outcome prediction [14, 30]. For instance, the studies in [31, 32, 23] have attempted to extract the TME information from WSIs for survival prediction. Han *et al.* [24] introduced a multi-scale heterogeneity-aware hypergraph representation framework to characterize the interactions between different TME components. Wu *et al.* [3] incorporated the concepts of prototype for TME analysis. Although these methods have shown promising results, they typically require dividing WSIs into patches, followed by training a dedicated segmentation or classification model, or fine-tuning a foundation model, to identify TME component types and select key patches before constructing graphs. However, such a patch pre-selection process relies heavily on additional annotations from pathologists to train a satisfactory classifier or segmentor, which is both labor-intensive and impractical when handling large volumes of high-resolution WSIs. What's more, they often neglect to incorporate semantic features during graph construction, limiting their ability to capture the biological significance of the interactions between different TME components.

## 3 Method

Fig. 1 presents the overview of our proposed ZTSurv. We leverage tumor infiltrating lymphocyte (TIL) maps on the 50x down-sampled WSIs to perform pixel-level zero-shot segmentation on other two important TME components (*i.e.,* tumor and stroma). The segmentation results are then used to construct a heterogeneous graph, where the different TME components are represented as nodes

based on their visual and semantic features. We capture the interactions between these components to form the graph edges. The constructed heterogeneous graph is updated through graph learning, which ultimately enables the prediction of survival outcomes.

## 3.1 Zero-shot TME Segmentation

Our objective is to segment TME tissue regions that are widely considered to be crucial for cancer prognosis (*e.g.,* lymphocyte [33], stroma [30], tumor [34]) on low-resolution WSIs using the corresponding TIL maps, where only the lymphocyte class is available as ground truth for training. To this end, we first down-sample the high-resolution WSI to match the size of the TIL map with width $W'$ and height $H'$. We employ the PLIP [35] vision encoder, which divides the image into $n$ patches with a patch size of $p'$, and incorporate a learnable vision prompt to extract visual representations, including the [cls] token $\boldsymbol{g} \in \mathbb{R}^{d'}$ and vision embedding $\boldsymbol{Z} = \{\boldsymbol{z}_1, \boldsymbol{z}_2, \ldots, \boldsymbol{z}_n\} \in \mathbb{R}^{n \times d'}$, where $d'$ is the feature dimension of the PLIP model. Specifically, the input embeddings from the $l$-th multi-head attention (MHA) module of the ViT-based encoder in PLIP are represented as $\{\boldsymbol{g}^l, \boldsymbol{z}_1^l, \boldsymbol{z}_2^l, \ldots, \boldsymbol{z}_n^l\}$. We add a learnable vision prompt $\boldsymbol{P}^l = \{\boldsymbol{p}_1^l, \boldsymbol{p}_2^l, \ldots, \boldsymbol{p}_m^l\}$ into the input and the $l$-th MHA module processes the tokens as follows (detailed illustrations are available in the *Appendix* C):

$$[\boldsymbol{g}^l, \boldsymbol{Z}^l, \_] = Layer^l[\boldsymbol{g}^{l-1}, \boldsymbol{Z}^{l-1}, \boldsymbol{P}^{l-1}]. \tag{1}$$

Instead of using generic text prompts like "a photo of {}" , we generate class-specific textual descriptions $\hat{\boldsymbol{H}}$ that capture the appearance attributes of various TME components using a large language model (LLM) (details in *Appendix* D). These descriptions are encoded via the PLIP text encoder to obtain text embeddings $\boldsymbol{H} \in \mathbb{R}^{C \times d'}$, where $C$ is the number of classes. A projector $\psi_p$, composed of three linear layers, maps the text embeddings to prototypes $\boldsymbol{E} = \psi_p(\boldsymbol{H}) \in \mathbb{R}^{C \times d'}$. Inspired by [36], we adopt a mask decoder consisting of three layers of lightweight transformers to segment the TME components at the pixel level. The input $\boldsymbol{Q}(query), \boldsymbol{K}(key), \boldsymbol{V}(value)$ are computed as:

$$\boldsymbol{Q} = \psi_q(\boldsymbol{E} \odot \boldsymbol{g}) \in \mathbb{R}^{C \times d'}, \quad \boldsymbol{K} = \psi_k(\boldsymbol{Z}) \in \mathbb{R}^{n \times d'}, \quad \boldsymbol{V} = \psi_v(\boldsymbol{Z}) \in \mathbb{R}^{n \times d'}, \tag{2}$$

where $\odot$ is the Hadamard product, $\psi_q$, $\psi_k$, and $\psi_v$ represent linear transformations. At each transformer layer, the query $\boldsymbol{Q}$ is updated to better capture the semantic correlations with the visual embeddings. To obtain the predicted mask, we calculate the score map using scaled dot-product attention from the final layer with a Sigmoid activation to ensure that the segmentation results of each class are independently generated:

$$\text{ScoreMap} = \text{Sigmoid}(\frac{\boldsymbol{Q}\boldsymbol{K}^T}{\sqrt{d_k}}) \in \mathbb{R}^{C \times n}, \tag{3}$$

where $\sqrt{d_k}$ is the dimension of the keys as a scaling factor. The score map is then reshaped to $C \times (H'/p') \times (W'/p')$, and the final segmentation mask $\boldsymbol{M} \in \mathbb{R}^{H' \times W'}$ is obtained by applying an Argmax operation along the class dimension followed by an up-sampling step to restore the original spatial resolution. We train the zero-shot segmentation model using focal loss [37] and dice loss [38, 39]. Notably, only the lymphocyte class, which is visible in the TIL map, contributes to the loss calculation, while other classes are ignored.

## 3.2 Heterogeneous Graph Construction

To explicitly model the heterogeneity and spatial organization of the TME, we construct a heterogeneous graph $\mathcal{G} = (\mathcal{V}, \mathcal{E}, \mathcal{A}, \mathcal{R})$ based on the predicted segmentation mask $\boldsymbol{M}$, where $\mathcal{V}, \mathcal{E}, \mathcal{A}, \mathcal{R}$ represent the set of entities (vertices or nodes), the set of relations (edges), the set of entity types, the space of edge attributes, respectively. Each node $v \in \mathcal{V}$ is associated with a type through a mapping function $\tau(v) \in \mathcal{A}$. An edge $e = (s, r, t) \in \mathcal{E}$ connects a source node $s$ to a target node $t$, where the edge type is given by $\phi(e) = r \in \mathcal{R}$. Each node $v$ has a $d$-dimensional feature vector $\boldsymbol{x} \in \mathcal{X}$, where $\mathcal{X}$ represents the embedding space for node feature. Specifically, we divide the down-sampled WSI into a set of non-overlapping patches using a fixed window size $s' \times s'$. For each patch, we determine its tissue label $y \in \mathcal{A}$ based on the dominant class within the patch area. Patches labeled

as background are discarded, and the remaining patches are retained as graph nodes $\mathcal{V}$. To construct node features $\mathcal{X}$, we first extract visual features $\boldsymbol{f}^{\text{vis}} \in \mathbb{R}^{d_v}$ for each patch using the PLIP vision encoder. In addition, considering the semantic gap and inherent heterogeneity among different TME components, we further obtain semantic embeddings $\boldsymbol{f}^{\text{text}} \in \mathbb{R}^{d_t}$ by feeding the corresponding class name into the PLIP text encoder. Then, the final node representation can be calculated as:

$$\boldsymbol{x} = [\boldsymbol{f}^{\text{vis}} \| \boldsymbol{f}^{\text{text}}] \in \mathbb{R}^{d_v+d_t}, \quad \boldsymbol{x} \in \mathcal{X}. \tag{4}$$

Here, $\|$ denotes the concatenation operation. The node types and node features reflect the biological roles of different TME components and preserve both visual and semantic heterogeneity at the node level. Based on the defined nodes and their feature representations, we establish edges and assign edge attributes to capture neighborhood relationships. For each node $v \in \mathcal{V}$, we apply the $k$-nearest neighbor algorithm to identify the $k$ most similar nodes to it, and create directed edges connecting $v$ to each of its neighbors. For each edge $e \in \mathcal{E}$, we compute the Pearson correlation coefficient $u \in \mathcal{U}$ between the feature vectors of the source and target nodes as its continuous attribute, where $\mathcal{U}$ represents the set of continuous edge attributes. The edge type $r \in \mathcal{R}$ is labeled as "positive" if the coefficient is positive and "negative" otherwise. The edge attributes introduce heterogeneity at the relational level and help to highlight implicit meta-relations among different tissue regions in the WSI. To reduce potential noise introduced by uncertain or spurious correlations, we apply data augmentation strategies during training, including random edge and feature dropout.

### 3.3 Heterogeneous Graph Learning for Survival Analysis

Traditional graph attention mechanisms fail to effectively address the heterogeneity inherent in the graph structure [3]. To address this challenge, we incorporate node features that capture both visual representations and type-specific information, alongside continuous edge attributes, into the aggregation process, which allows the model to capture the complex interactions and diverse relationships between different TME components.

**Edge Updating.** For each edge $e \in \mathcal{E}$, we project its continuous attributes $u^{l-1} \in \mathcal{U}$ from the $(l-1)$-th graph learning layer to the $l$-th layer $u^l = W_{edge} u^{l-1}$ using a linear projector $W_{edge}$.

**Node Updating.** Instead of using edge similarity as weights for node updates, we account for the heterogeneity inherent in both edges and nodes (detailed illustrations are available in the *Appendix* F). Specifically, in each layer, we update the embedding of a node $v \in \mathcal{V}$ by aggregating information from all its neighbors. We refer to node $v$ as target node $t$, with its neighbors denoted as $\mathcal{V}(t) = \{s_1, s_2, \ldots, s_N\}$, where $\mathcal{V}(t)$ represents the set of source nodes that point to the target node $t$, and $N$ is the number of neighbors. The set of edges associated with node $t$ is denoted as $\mathcal{E}(t) = \{e_1, e_2, \ldots, e_N\}$. At layer $l$, for each $(s_i, e_i, t)$ and attention head $h$, we first project the target node $t$ into a query vector $\boldsymbol{F}^h_{\text{query}}$ using a linear projector $\boldsymbol{W}^h_{\tau(t)}$, and the source node into key and value vectors $\boldsymbol{F}^h_{\text{key},i}$ and $\boldsymbol{F}^h_{\text{value},i}$ using $\boldsymbol{W}^h_{\tau(s_i)}$:

$$\boldsymbol{F}^h_{\text{query}} = \boldsymbol{W}^h_{\tau(t)} \boldsymbol{x}^{l-1}_t, \quad \boldsymbol{F}^h_{\text{key},i} = \boldsymbol{W}^h_{\tau(s_i)} \boldsymbol{x}^{l-1}_{s_i}, \quad \boldsymbol{F}^h_{\text{value},i} = \boldsymbol{W}^h_{\tau(s_i)} \boldsymbol{x}^{l-1}_{s_i}, \tag{5}$$

where $\boldsymbol{x}^{l-1}_{s_i}$ and $\boldsymbol{x}^{l-1}_t$ represent the node features of the source node $s_i$ and target node $t$ at the $(l-1)$-th layer, and $\tau(\cdot) \in \mathcal{A}$ is a mapping function that assigns the corresponding type to each node. Then we calculate the attention score for each edge $e_i$ on $h$-th attention head using the query vector $\boldsymbol{F}^h_{\text{query}}$ and the key vector $\boldsymbol{F}^h_{\text{key},i}$ modulated by the edge's continuous attribute $u_i$, and apply the Softmax function across all edges in $\mathcal{E}(t)$ to obtain the normalized attention scores:

$$att^h(t, e_i) = \boldsymbol{F}^h_{\text{key},i} \cdot u_i \cdot \boldsymbol{F}^h_{\text{query}} / \sqrt{d_v + d_t}, \tag{6}$$

$$w^h(t, e) = \underset{\forall e \in \mathcal{E}(t)}{\text{Softmax}}(att^h(t, e)), \tag{7}$$

where $\sqrt{d_v + d_t}$ is the scaling factor to ensure numerical stability, $w^h(t, e) \in \mathbb{R}^N$ represents the final attention score of the edges associated with target node $t$ on $h$-th attention head. Finally, the updated

embedding of the target node $t$ is computed by aggregating the weighted value vectors:

$$\boldsymbol{x}_t = \sum_{i=1}^{N}(\|_{h\in[1,H]} w^h(t, e_i) \cdot \boldsymbol{F}^h_{\text{value},i}), \tag{8}$$

where $\|_{h\in[1,H]}$ denotes the concatenation of all attention heads. The aggregation process results in an updated feature that effectively combines both node-type and edge-attribute information, thereby capturing the graph's heterogeneity. After completing the $L$-th layer of graph learning, we employ a global attention based pooling [40] to dynamically calculate a weighted sum of the node features in the graph, transforming them into a WSI-level embedding representation. The representation is then passed through a MLP to predict the final survival risk score. For the $k'$-th patient, we can model the survival function $f^{(k')}_{surv}(T \geq t, D^{(k')})$ and hazard function $f^{(k')}_{hazard}(T = t|T \geq t, D^{(k')})$ given the relative clinical information $D^{(k')} = (X^{(k')}, c^{(k')}, t^{(k')})$, where $X^{(k')}$ represents the patient's WSI, $c^{(k')} \in \{0,1\}$ indicates the censoring status, and $t^{(k')} \in \mathbb{R}^+$ denotes the overall survival time. After graph learning, we learn the representation $f(D^{(k')})$ and compute the survival loss $\mathcal{L}_{surv}(\{f(D^{(k')}), t^{(k')}, c^{(k')}\}^{N_D}_{k'=1})$ of all patients, where $N_D$ is the number of samples in the training set. Specifically, we adopt the negative log-likelihood (NLL) loss [41] to quantify the difference between the predicted survival risk and the actual clinical outcomes (details can be found in the *Appendix* B):

$$\mathcal{L}_{surv} = -\sum_{k'=1}^{N_D} c^{(k')} log(f^{(k')}_{surv}(t|f(D^{(k')}))) \tag{9}$$

$$+ (1 - c^{(k')}) log(f^{(k')}_{surv}(t-1|f(D^{(k')}))) \tag{10}$$

$$+ (1 - c^{(k')}) log(f^{(k')}_{hazard}(t|f(D^{(k')}))). \tag{11}$$

### 3.4 Overall Loss

The overall loss consists of the zero-shot TME segmentation loss $\mathcal{L}_{seg}$ and the survival loss $\mathcal{L}_{surv}$, which can be formulated as follows:

$$\mathcal{L} = \mathcal{L}_{seg} + \gamma\mathcal{L}_{surv} = \alpha\mathcal{L}_{focal} + \beta\mathcal{L}_{dice} + \gamma\mathcal{L}_{surv}, \tag{12}$$

where $\{\alpha, \beta, \gamma\}$ are coefficients that balance the contributions of different losses. In *Appendix* H, we further analysis the time complexity of ZTSurv.

## 4 Experiments

### 4.1 Datasets

We conduct experiments on four cancer cohorts: Breast Invasive Carcinoma (BRCA) (1,016 cases), Uterine Corpus Endometrial Carcinoma (UCEC) (520 cases), Lung Adenocarcinoma (LUAD) (540 cases), and Bladder Urothelial Carcinoma (BLCA) (369 cases). All WSIs for these cancer types are sourced from The Cancer Genome Atlas (TCGA) repository [42] [2]. The corresponding 50x down-sampled tumor-infiltrating lymphocyte (TIL) maps are obtained from [43] [3].

### 4.2 Experimental Settings

**Implementation Details.** For each cancer cohort, we evaluate model performance using a 5-fold cross-validation strategy. We use openslide [44] tool to process and down-sample the high-resolution WSIs. The WSI segmentation is implemented with the open-source MMSegmentation toolbox [45] with PyTorch 1.10.1. We employ the pre-trained PLIP ViT-B/32 model to extract visual and textual features, with a feature dimension of 512. GPT-4 [46] is used as the LLM to generate class

---

[2]https://portal.gdc.cancer.gov/
[3]https://www.cancerimagingarchive.net/analysis-result/til-wsi-tcga/

Table 1: Comparisons of C-index (mean ± std) for survival prediction with SOTA methods over four cancer datasets. The best and the second-best results are highlighted in **bold** and underlined. p. represents methods sampling patches from WSIs; g. denotes graph-based methods; t. refers to methods considering TME heterogeneity; w. represents methods without splitting WSIs into patches.

| Model | Design | BRCA | UCEC | LUAD | BLCA | Overall |
|---|---|---|---|---|---|---|
| CLAM-SB [51] | p. | 0.581 ± 0.041 | 0.550 ± 0.113 | 0.549 ± 0.053 | 0.563 ± 0.049 | 0.561 |
| CLAM-MB [51] | p. | 0.541 ± 0.119 | 0.551 ± 0.118 | 0.556 ± 0.067 | 0.601 ± 0.070 | 0.562 |
| Co-Pilot [28] | p. | 0.544 ± 0.076 | 0.557 ± 0.112 | 0.571 ± 0.083 | 0.587 ± 0.056 | 0.564 |
| DeepAttnMISL [12] | p. | 0.598 ± 0.066 | 0.639 ± 0.068 | 0.601 ± 0.041 | 0.597 ± 0.028 | 0.609 |
| DSMIL [11] | p. | 0.548 ± 0.080 | 0.581 ± 0.164 | 0.538 ± 0.047 | 0.552 ± 0.052 | 0.555 |
| WSISA [6] | p. | 0.514 ± 0.071 | 0.539 ± 0.114 | 0.575 ± 0.055 | 0.581 ± 0.050 | 0.552 |
| DeepGraphSurv [21] | p.+g. | 0.587 ± 0.033 | 0.622 ± 0.097 | 0.595 ± 0.011 | 0.584 ± 0.037 | 0.597 |
| HGSurvNet [22] | p.+g. | 0.624 ± 0.093 | 0.614 ± 0.034 | 0.587 ± 0.045 | 0.578 ± 0.041 | 0.601 |
| PatchGCN [16] | p.+g. | 0.608 ± 0.043 | 0.678 ± 0.128 | 0.614 ± 0.007 | 0.599 ± 0.034 | 0.625 |
| H2GT [24] | p.+g.+t. | 0.618 ± 0.032 | 0.671 ± 0.096 | 0.629 ± 0.046 | 0.623 ± 0.058 | 0.635 |
| TMEGL [2] | p.+g.+t. | 0.623 ± 0.017 | 0.700 ± 0.053 | 0.631 ± 0.023 | 0.627 ± 0.029 | 0.645 |
| ProtoSurv [3] | p.+g.+t. | 0.625 ± 0.009 | 0.705 ± 0.131 | **0.638 ± 0.026** | 0.629 ± 0.043 | 0.649 |
| ZTSurv (ours) | w.+g.+t. | **0.642 ± 0.029** | **0.726 ± 0.113** | 0.637 ± 0.033 | **0.637 ± 0.042** | **0.661** |

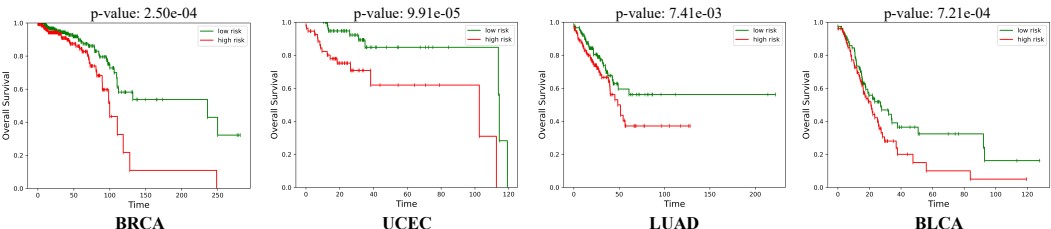

Figure 2: Kaplan–Meier curves for predicted high-risk (red) and low-risk (green) groups across four cancer datasets. A p-value $< 0.05$ indicates statistical significance.

descriptions for different tissue categories. For graph construction, we set the window size $s' = 64$ and select $k = 8$ neighbors per node. Data augmentation is applied by randomly dropping node and edge features with a dropout rate of 0.2. The layer for graph learning $L$ is set to 2, with $H = 4$ attention heads. The hyperparameters for the loss function are set as $\alpha = 20$, $\beta = 1$, and $\gamma = 20$. We use the AdamW optimizer [47] with a learning rate of $2 \times 10^{-5}$ and a weight decay of $1 \times 10^{-5}$. The batch size is set to 8, and we train the model for 2K iterations. A detailed parameter analysis is provided in the *Appendix* H.5.

**Evaluation Metrics.** To evaluate TME segmentation performance, we use the mean of class-wise Intersection over Union (mIoU) on lymphocyte, *i.e.,* the only TME component with available ground truth annotations, to measure the overlap between predicted and reference regions. To assess predictive performance, we use the concordance index (C-index) [48] for evaluating the ability to correctly rank the survival risk of different patients. For qualitative assessment, we employ Kaplan-Meier curves [49] with log-rank test [50] to visualize patient stratification, distinguishing between low and high-risk patients with two separate survival distributions. More details can be found in *Appendix* G.

### 4.3 Comparison with State-Of-The-Art Methods.

We compare our proposed method with several state-of-the-art approaches for survival prediction: (1) CLAM-SB [51], (2) CLAM-MB [51], (3) Co-Pilot [28], (4) DeepAttnMISL [12], (5) DSMIL [11], (6) WSISA [6], (7) DeepGraphSurv [21], (8) HGSurvNet [22], (9) PatchGCN [16], (10) H2GT [24], (11) TMEGL [2], (12) ProtoSurv [3]. Among them, CLAM-SB, CLAM-MB, Co-Pilot, DeepAttnMISL, DSMIL, and WSISA are classical WSI survival prediction approaches; DeepGraphSurv, HGSurvNet, and PatchGCN model spatial relationships by constructing homogeneous graphs for survival prediction; H2GT, TMEGL, and ProtoSurv leverage the heterogeneity of the TME for survival analysis. All

Table 2: Comparisons of mIoU (mean ± std) for TME segmentation with SOTA methods over four cancer datasets. The best and the second-best results are highlighted in **bold** and underlined.

| Model | BRCA | UCEC | LUAD | BLCA | Overall |
|-------|------|------|------|------|---------|
| ZegFormer [52] | $0.536 \pm 0.109$ | $0.459 \pm 0.125$ | $0.566 \pm 0.021$ | $0.561 \pm 0.047$ | 0.531 |
| ZegCLIP [53] | $0.547 \pm 0.105$ | $0.475 \pm 0.121$ | $0.572 \pm 0.040$ | $0.570 \pm 0.055$ | 0.541 |
| TagCLIP [54] | $0.541 \pm 0.117$ | $0.474 \pm 0.141$ | $0.568 \pm 0.036$ | $0.579 \pm 0.059$ | 0.541 |
| ZTSurv-Seg | $0.552 \pm 0.096$ | $0.478 \pm 0.150$ | $0.579 \pm 0.050$ | $0.589 \pm 0.056$ | 0.550 |
| ZTSurv (ours) | $\mathbf{0.574 \pm 0.023}$ | $\mathbf{0.506 \pm 0.147}$ | $\mathbf{0.595 \pm 0.023}$ | $\mathbf{0.608 \pm 0.055}$ | **0.571** |

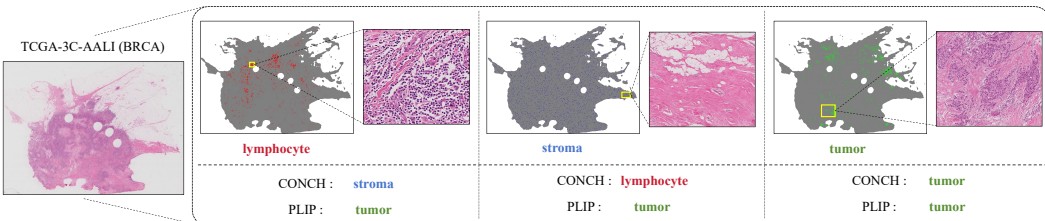

Figure 3: Zero-shot TME segmentation comparison of ZTSurv, CONCH, and PLIP.

these methods require splitting WSIs into patches as an essential processing step, while ZTSurv can avoid this step and directly utilize low-resolution WSIs for training an end-to-end survival prediction model.

As shown in Table 1, ZTSurv outperforms all classical [51, 28, 12, 11, 6] and graph-based [21, 22, 16] methods by a margin of 1.8%-4.8%, demonstrating the effectiveness of our method that incorporates TME heterogeneity for more accurate survival prediction. Compared to recent TME-aware models such as H2GT [24], TMEGL [2], and ProtoSurv [3], ZTSurv still achieves the highest C-index on 3 out of 4 cancer datasets, with an improvement of 0.8%-2.1%. These results highlight the strength of ZTSurv that can simultaneously realize the TME segmentation and survival prediction tasks, while the existing studies treat these two tasks independently that miss the inherent correlation among them.

## 4.4 Patient Stratification

We perform Kaplan–Meier analysis by separating patients into high-risk and low-risk groups based on predicted risk scores, using the median value within each validation set as the cut-off. The log-rank test [50] is then applied to compute p-values that assess the statistical significance of survival differences between the two groups, with smaller p-values indicating better stratification. In Fig. 2 and *Appendix* H.1, we present the stratification ability of our method across four cancer datasets and compare it with two best competitors (*i.e.,* ProtoSurv and TMEGL). The results clearly show that our method achieves more distinct separation between high-risk (red) and low-risk (green) groups across all cohorts with lower p-values, which further demonstrates the effectiveness of our method.

## 4.5 Ablation Study and Analysis

**Component Ablation.** To further investigate the contribution of each component in ZTSurv, we conduct an ablation study by removing or replacing key modules in *Appendix* H.2: 1) w/o text prompt: Removing text prompt of zero-shot segmentation. 2) w/o vision prompt: Removing vision prompt of zero-shot segmentation. 3) w/o node type: Constructing a homogeneous graph without considering node types during graph construction. 4) w/o semantic feature: Without considering semantic feature for node representation during graph construction. 5) w/o vision feature: Without considering vision feature for node representation during graph construction. 6) w/o edge attribute: Without considering edge attribute during graph construction and learning. As can be observed from Table 4 in *Appendix*, ZTSurv is superior to its variants, indicating that each component of our method is effective in improving survival prediction performance.

**Comparison of Zero-shot TME Segmentation.** We compare the zero-shot TME segmentation performance with several state-of-the-art methods: (1) ZegFormer [52], (2) ZegCLIP [53], (3)

Table 3: Effect of tissue category choices on C-index (mean ± std) over four cancer datasets.

|  | BRCA | UCEC | LUAD | BLCA | Overall |
|---|---|---|---|---|---|
| lymphocyte + tumor | 0.628 ± 0.020 | 0.713 ± 0.128 | 0.629 ± 0.06 | 0.629 ± 0.047 | 0.650 |
| lymphocyte + stroma | 0.634 ± 0.043 | 0.704 ± 0.056 | 0.633 ± 0.065 | 0.623 ± 0.049 | 0.649 |
| lymphocyte + tumor + stroma | 0.642 ± 0.029 | 0.726 ± 0.113 | 0.637 ± 0.033 | 0.637 ± 0.042 | 0.661 |

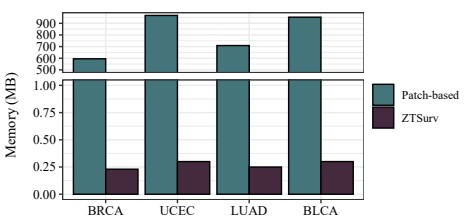

(a) Average Memory Usage per WSI.

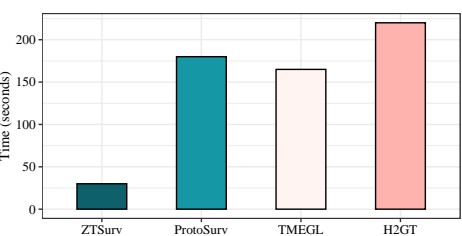

(b) Average Inference Time per WSI.

Figure 4: Comparison of Average Memory Usage and Inference Time per WSI.

TagCLIP [54]. We also evaluate a variant of our method, ZTSurv-seg, which is trained solely with segmentation loss and without survival-guided information. As shown in Table 2, ZTSurv consistently outperforms all compared methods across four cancer datasets, achieving the highest overall mIoU of 0.571. Additionally, compared to its variant ZTSurv-seg, which lacks survival guidance, ZTSurv shows consistent gains in segmentation performance across all cohorts. These results suggest that incorporating survival supervision can provide informative signals that help refine the segmentation process, resulting in more prognostically meaningful TME delineation. In Fig. 3, we further present the visualization results of segmentation and compare it with CONCH [55] and PLIP [35]. As shown in Fig. 3, our method accurately captures key TME components, while existing survival analysis approaches that rely on pathology foundation models, like CONCH and PLIP, often struggle to identify non-tumor components, limiting their ability to fully capture TME heterogeneity. The results further highlight the effectiveness of our approach in zero-shot TME segmentation, enabling more comprehensive and precise survival analysis. In the *Appendix* H.4, we provide more visualization results on other three datasets (*i.e.,* BLCA, LUAD, and UCEC).

**Comparison of Survival Prediction with Different Zero-shot Classifiers.** In *Appendix* H.3, we compare ZTSurv with three alternative approaches for the zero-shot TME segmentation stage (*i.e.,* CONCH [55], PLIP [35], and a UNI classifier [56] finetuned as described in [3]) for survival outcome prediction. As shown in Fig. 9 in *Appendix*, ZTSurv consistently achieves superior C-index scores across all four cancer datasets, demonstrating that our method achieves better performance.

**Effect of Different Tissue Categories.** In our work, we consider three key TME components, (*i.e.,* lymphocyte, tumor, and stroma), to capture the complex TME. To further investigate the impact of different tissue categories on survival prediction, we conduct experiments with different combinations of tissue components (*i.e.,* lymphocyte + tumor, lymphocyte + stroma) to evaluate their influence on survival analysis. As shown in Table 3, including a more comprehensive set of tissue categories (*i.e.,* lymphocyte + tumor + stroma) consistently achieves the highest C-index across four datasets, indicating that capturing a broader TME leads to better survival prediction.

**Time and Memory Analysis.** Compared with conventional patch-based methods that divide high-resolution WSIs into thousands of tiles, ZTSurv offers significant advantages in both time and memory efficiency. As shown in Fig. 4, on a single NVIDIA RTX 4090 GPU, ZTSurv reduces average memory usage per WSI by approximately 3,000 times and decreases average inference time by about 6 times compared to other methods. This is primarily because ZTSurv directly processes the down-sampled WSI without the need to generate and store thousands of small patches, significantly reducing computational overhead and memory requirements, which further demonstrates its superior scalability and efficiency for large-scale whole-slide image analysis.

# 5 Conclusion

In this paper, we propose ZTSurv, a novel end-to-end framework for cancer survival analysis that integrates survival prediction with zero-shot TME segmentation on low-resolution WSIs. ZTSurv eliminates the need for manual annotations by leveraging TIL maps to segment key TME components, and introduces survival-guided segmentation to enhance the identification of prognostically TME regions. By performing segmentation directly on 50× down-sampled WSIs, the framework significantly reduces computational cost. Furthermore, the incorporation of semantic features in graph construction allows the model to better capture complex tissue interactions. Extensive experiments on four TCGA cohorts demonstrate the effectiveness and efficiency of ZTSurv, indicating its strong potential for scalable and practical histopathology analysis.

## Acknowledgment

This work was supported by the National Natural Science Foundation of China (Nos. 62136004, 62272226, 62102188), Key Research and Development Plan of Jiangsu Province, China under Grant BE2022842, National Key R&D Program of China (Grant No. 2023YFF1204803).

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

## A Overview

In this appendix, we first provide detailed analysis of survival prediction in B. We then provide the detailed illustrations of vision prompt and text prompt for zero-shot TME segmentation in C and D, respectively. In E and F, we present the detailed illustrations of mask decoder of zero-shot TME segmentation and node updating in graph learning, respectively. Then, we provide the evaluation metric for survival prediction in G. In H, we present more experiments and analysis, including comparisons of patient stratification H.1, ablation study of main components H.2, comparisons of survival prediction with different zero-shot classifiers H.3, more zero-shot TME segmentation visualization results H.4, parameter analysis H.5, time complexity analysis H.6, and discussions of limitations and future work H.7. Finally, we discuss the potential ethical issues which may arise in our study in I.

## B Survival Analysis

Survival prediction aims to model the time until an event occurs, typically formulated as an ordinal regression problem. In clinical data, the event (*e.g.,* death) is not always observed due to censoring, such as when a patient is lost to follow-up. These right-censored cases introduce uncertainty, as we only know that the event did not occur before the last recorded time.

Following our notation in Sec. 3.3, let $D = (X, c, t)$ represent the patient's clinical data, where $X$ is patient's pathology image, $t \in \mathbb{R}^+$ denotes the overall survival time, $c \in \{0, 1\}$ specifies whether the data is right-censored. We denote $T$ as a continuous random variable representing survival time. The survival function $f_{surv}(T \geq t, D)$ estimates the probability of a patient surviving beyond time $t$, while the hazard function $f_{hazard}(t|D) = f_{hazard}(T = t|T \geq t, D)$ quantifies the risk of the event occurring at time $t$ given survival up to that point, which is defined as:

$$f_{hazard}(T = t) = \lim_{\partial t \to 0} \frac{P(t \leq T \leq t + \partial t \mid T \geq t)}{\partial t}, \tag{13}$$

which can be used to estimate $f_{surv}^{(k')}(T \geq t, D^{(k')})$ by integrating the hazard function $f_{hazard}$. The most common model for learning hazard functions is the Cox Proportional Hazards (CoxPH) model, where the hazard is modeled as:

$$f_{hazard}(t|D) = f_{hazard}^0(t) \exp(\theta^\top D). \tag{14}$$

Here, $f_{hazard}^0(t)$ is the baseline hazard, and $\theta$ contains parameters that modulate risk based on the input features $D$. In deep learning applications, $\theta$ is typically produced by the final layer of a neural network and is optimized using the partial log-likelihood of the Cox model via stochastic gradient descent.

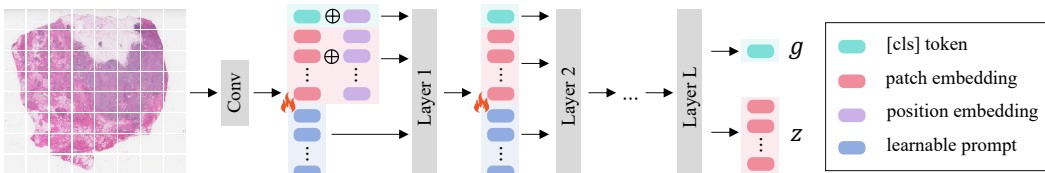

Figure 5: Illustration of vision prompt for zero-shot TME segmentation.

## C Vision Prompt of Zero-shot TME Segmentation

In Sec. 3.1, we use the vision prompt, as shown in Fig. 5, to enhance the extraction of visual features from low-resolution WSIs. The [cls] token, patch embeddings, and learnable prompts are passed through each layer, with only the learnable prompts being trainable.

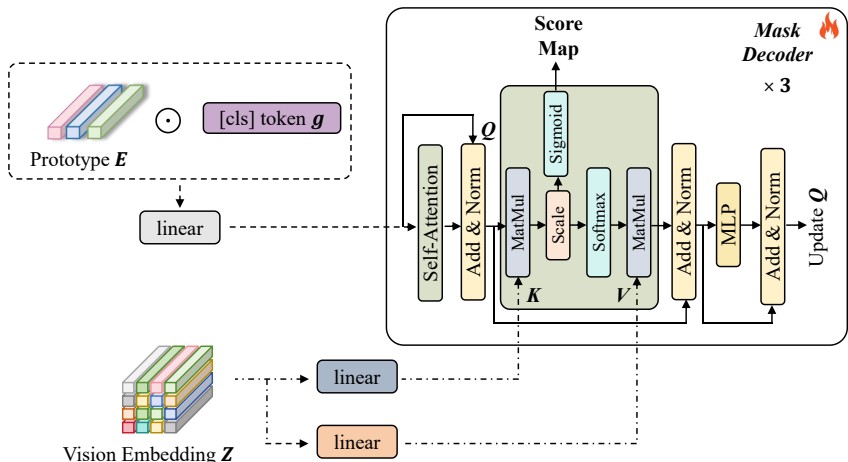

Figure 6: A detailed illustration of mask decoder in zero-shot TME segmentation.

## D    Text Prompt of Zero-shot TME Segmentation

In Sec. 3.1, we generate class-specific textual descriptions which capture the appearance attributes of TME components using the LLM. Specifically, instead of using generic prompts like *"a photo of {}"*, we use the following prompts fed into the LLM: *"Describe {} in the pathology image in detail, including features such as color and shape"*. We list the textual descriptions for key TME components (*i.e.,* lymphocyte, stroma, tumor) as follows:

- "lymphocyte": This is the lymphocyte in the pathology image, typically appearing as small, round cells with a light blue cytoplasm and dark purple nuclei.
- "stroma": This is the stroma in the pathology image, typically appearing as fibrous tissue in pale pink or beige tones, supporting the surrounding cells.
- "tumor": This is the tumor in the pathology image, typically appearing as irregularly shaped cells with darker purple or blue cytoplasm and enlarged, pleomorphic nuclei.

## E    Detailed Illustrations of Mask Decoder

In Fig. 6, we present a detailed illustration of the mask decoder inspired by [36] in zero-shot TME segmentation, which progressively refines feature representations to generate accurate segmentation masks.

## F    Detailed Illustrations of Node Updating

In Fig. 7, we provide a detailed illustration of node updating in graph learning, where the target node is solely associated with source nodes $A$ and $B$ in this example.

## G    Evaluation Metric for Survival Analysis

We use the concordance index (C-index) to evaluate the predictive accuracy of survival analysis models, which evaluates a model's ability to correctly rank pairs of samples based on their predicted risk of experiencing an event (*e.g.,* death) within a given time frame. Specifically, samples are sorted by their predicted survival scores, and the C-index reflects the proportion of correctly ordered pairs. The metric is defined as:

$$C\text{-}index = \frac{1}{N'(N'-1)} \sum_{i=1}^{N} \sum_{j=1}^{N} \mathbb{I}(T_i < T_j)(1 - c_j), \tag{15}$$

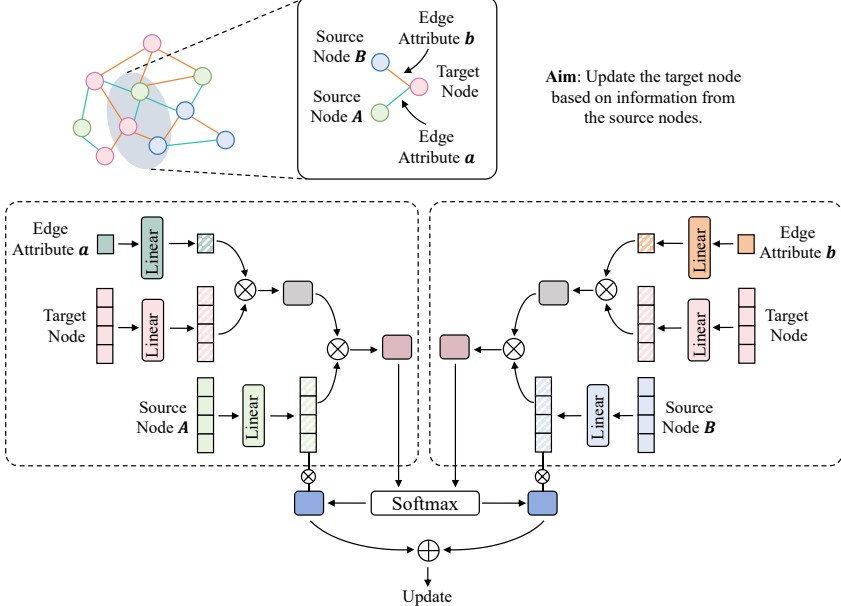

Figure 7: A detailed illustration of node updating in graph learning.

where $N'$ is the total number of patients, and $\mathbb{I}$ is the indicator function that returns 1 when the condition holds and 0 otherwise.

# H More Experiments and Analysis

## H.1 Patient Stratification

In addition to evaluating prognostic performance using the C-index, patient stratification is another key aspect of cancer survival analysis, enabling the identification of subgroups with distinct clinical outcomes for personalized treatment. We compare the patient stratification ability of ZTSurv with its two best competitors (*i.e.,* ProtoSurv and TMEGL) in Fig. 8, and the results demonstrate that ZTSurv consistently achieves clearer separation between risk groups, indicating more accurate prognosis.

## H.2 Component Ablation

Table 4 shows the ablation results of ZTSurv on C-index across four cancer datasets. As shown in Table 4, removing either the text prompt or vision prompt in TME segmentation results in a noticeable drop in performance across all datasets, with the overall C-index decreasing to 0.644 and 0.650 respectively, confirming the effectiveness of prompt-based guidance for zero-shot segmentation. In terms of graph construction, we further examine the influence of node types and feature representations. It is clear that removing node type information, *i.e.,* treating all nodes as homogeneous, leads to a noticeable performance drop, underscoring the importance of preserving semantic distinctions among TME regions. Furthermore, excluding the vision feature results in the most significant degradation, indicating that visual cues derived from WSIs are critical for accurate survival prediction. The absence of semantic features also degrades performance, suggesting their complementary value in capturing contextual information. Additionally, eliminating edge attributes weakens the model's ability to capture spatial relationships, resulting in a further drop in the overall C-index to 0.651. The results emphasize the critical role of each component in our framework.

## H.3 Comparison of Survival Prediction with Different Zero-shot Classifiers

In Fig. 9, we compare ZTSurv with three alternative approaches for the zero-shot TME segmentation stage (*i.e.,* CONCH [55], PLIP [35], and a UNI classifier [56] finetuned as described in [3]) to predict

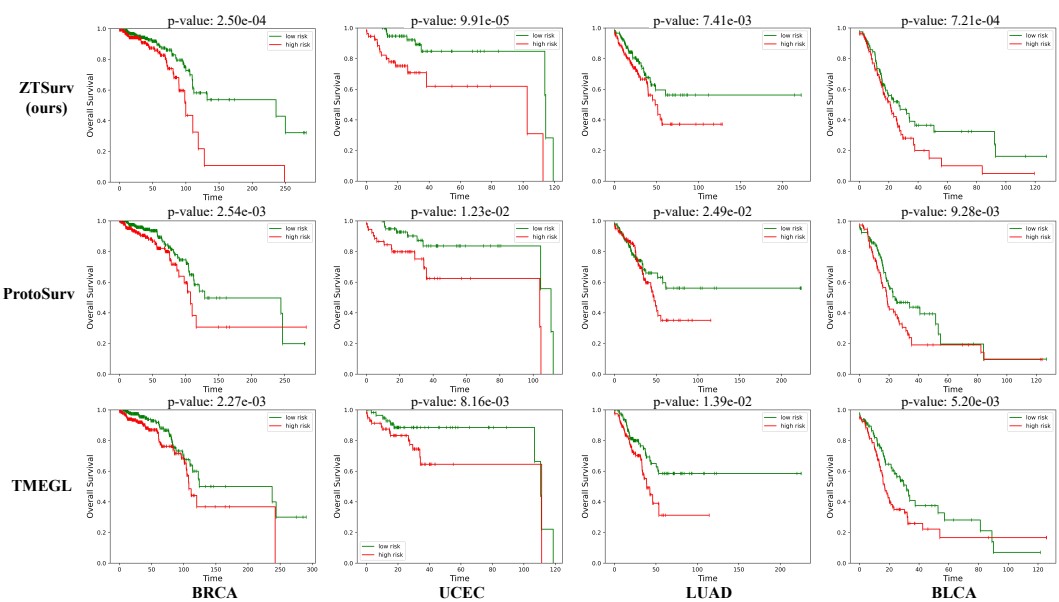

Figure 8: Kaplan–Meier curves for predicted high-risk (red) and low-risk (green) groups across four cancer datasets under different comparative methods. A p-value < 0.05 indicates statistical significance.

Table 4: Ablation study of ZTSurv on C-index (mean ± std) over four cancer datasets.

|  | **BRCA** | **UCEC** | **LUAD** | **BLCA** | **Overall** |
|---|---|---|---|---|---|
| w/o text prompt | $0.618 \pm 0.021$ | $0.706 \pm 0.098$ | $0.631 \pm 0.035$ | $0.622 \pm 0.018$ | 0.644 |
| w/o vision prompt | $0.621 \pm 0.016$ | $0.723 \pm 0.089$ | $0.623 \pm 0.026$ | $0.631 \pm 0.027$ | 0.650 |
| w/o node type | $0.615 \pm 0.025$ | $0.702 \pm 0.110$ | $0.624 \pm 0.019$ | $0.630 \pm 0.043$ | 0.643 |
| w/o semantic feature | $0.624 \pm 0.023$ | $0.701 \pm 0.071$ | $0.635 \pm 0.015$ | $0.624 \pm 0.026$ | 0.646 |
| w/o vision feature | $0.611 \pm 0.026$ | $0.669 \pm 0.075$ | $0.625 \pm 0.040$ | $0.621 \pm 0.023$ | 0.632 |
| w/o edge attribute | $0.632 \pm 0.024$ | $0.707 \pm 0.072$ | $0.632 \pm 0.036$ | $0.632 \pm 0.006$ | 0.651 |
| ZTSurv | $0.642 \pm 0.029$ | $0.726 \pm 0.113$ | $0.637 \pm 0.033$ | $0.637 \pm 0.042$ | 0.661 |

the survival outcome. The results further demonstrate the effectiveness of our method in capturing TME heterogeneity.

### H.4 More Zero-shot TME Segmentation Visualization

In Fig. 10, we provide more zero-shot TME segmentation visualization results on BLCA, LUAD, and UCEC cancer datasets. The results demonstrate that ZTSurv can capture key TME components more accurately.

### H.5 Parameter Analysis

We analyze the sensitivity of the number of neighbors $k$ used for connecting nodes and the window size $s'$ for node generation. As shown in Table 5, $k = 8$ yields the best overall performance, while both smaller ($k = 4$) and larger ($k = 16$) values result in performance degradation. For the window size $s'$, we observe that $s' = 64$ achieves the highest C-index across all datasets, while smaller window sizes result in overly complex graphs with redundant information and larger sizes fail to capture sufficient local detail, which highlights the importance of balancing graph density and information granularity in node construction.

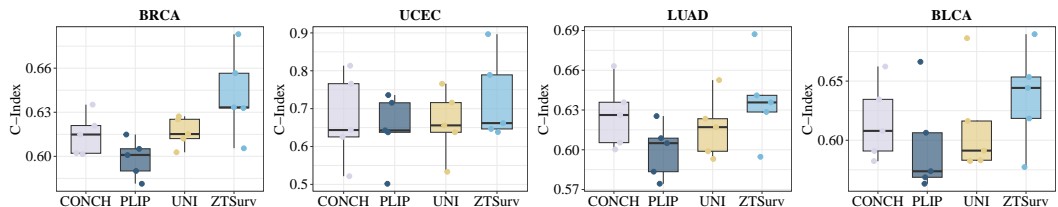

Figure 9: Comparisons of C-index (mean $\pm$ std) with different TME segmentation methods over four cancer datasets.

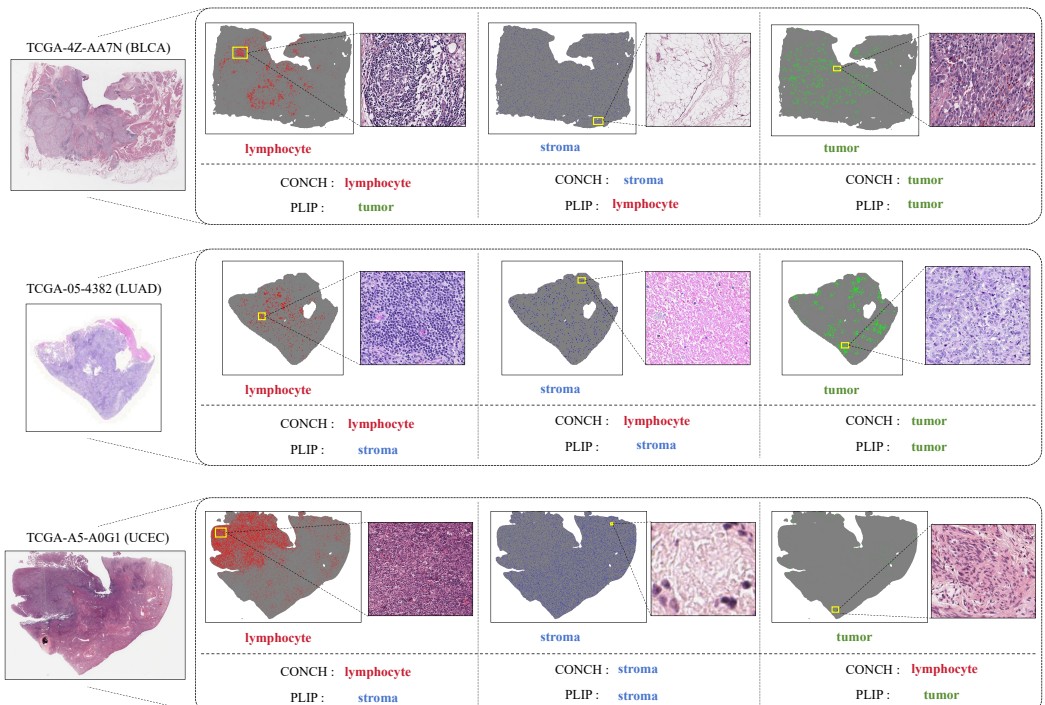

Figure 10: Zero-shot TME segmentation comparison of ZTSurv, CONCH, and PLIP.

## H.6 Time Complexity Analysis

The complexity of ZTSurv is dominated by three components. Firstly, zero-shot TME segmentation on the down-sampled WSI has a complexity of $O(H'W'd')$. Secondly, the complexity of constructing a heterogeneous graph with $n'$ nodes is $O(n'(d_v + d_t))$. Finally, the graph neural network performs message passing over $L$ layers, with a per-layer cost of $O(n'k(d_v + d_t))$. In summary, the time complexity of our ZTSurv is $O(H'W'd') + O(n'(d_v + d_t)) + O(Ln'k(d_v + d_t)) = O(H'W'd' + n'(1 + Lk)(d_v + d_t))$.

## H.7 Limitations and Future Work

In this work, we focus on capturing key TME components, including lymphocyte, tumor, and stroma, for survival prediction. However, as demonstrated in Table 3, including a broader range of TME components can potentially lead to better predictive performance. In future work, we will explore more diverse tissue types, such as blood vessels, necrosis, and fibroblasts, to capture a more comprehensive representation of the TME and further enhance model robustness.

Table 5: Parameter analysis of C-index (mean $\pm$ std) over four cancer datasets.

|  | BRCA | UCEC | LUAD | BLCA | Overall |
|---|---|---|---|---|---|
| $k = 4$ | $0.633 \pm 0.037$ | $0.703 \pm 0.051$ | $0.636 \pm 0.047$ | $0.632 \pm 0.045$ | 0.651 |
| $k = 8$ | $0.642 \pm 0.029$ | $0.726 \pm 0.113$ | $0.637 \pm 0.033$ | $0.637 \pm 0.042$ | 0.661 |
| $k = 16$ | $0.630 \pm 0.022$ | $0.698 \pm 0.063$ | $0.622 \pm 0.049$ | $0.625 \pm 0.041$ | 0.644 |
| $s' = 32$ | $0.635 \pm 0.022$ | $0.702 \pm 0.073$ | $0.632 \pm 0.052$ | $0.634 \pm 0.057$ | 0.651 |
| $s' = 64$ | $0.642 \pm 0.029$ | $0.726 \pm 0.113$ | $0.637 \pm 0.033$ | $0.637 \pm 0.042$ | 0.661 |
| $s' = 128$ | $0.635 \pm 0.037$ | $0.695 \pm 0.038$ | $0.631 \pm 0.037$ | $0.611 \pm 0.024$ | 0.643 |

# I Ethical Discussions

**Ethical Considerations.**    The Cancer Genome Atlas (TCGA) dataset used in this study is a publicly available resource widely utilized in pathology research. Given its open nature and established use, its application in this study does not present significant ethical concerns. The TIL maps employed were generated based on publicly available data without the involvement of any identifiable patient information, ensuring no individuals are adversely impacted. As such, this study adheres to ethical guidelines without compromising privacy or rights.

**Potential Positive Social Impacts.**    The proposed method has the potential to improve patient outcomes by enabling more accurate and comprehensive analysis of tumor microenvironments, facilitating early diagnosis and personalized treatment planning. Moreover, it can reduce the workload of pathologists by automating routine analyses, potentially increasing the scalability and efficiency of cancer diagnosis in clinical practice.

**Potential Negative Social Impacts.**    As this work focuses on cancer survival prediction, it is important to acknowledge potential social impacts, including but not limited to:

- Diagnostic Errors. Like all AI-based methods, this approach is not immune to errors. Incorrect predictions or misclassifications could have serious consequences for patient care and treatment decisions. Therefore, these tools should serve as decision aids, complementing but not replacing human medical judgment.
- Privacy Concerns. WSI datasets can contain sensitive information, and the leakage of such data may pose significant privacy risks to patients. To mitigate this, our study exclusively relies on publicly available datasets where personal identifiers are either absent or appropriately protected.

