# OpenReview forum: "Cancer Survival Analysis via Zero-shot Tumor Microenvironment Segmentation on Low-resolution Whole Slide Pathology Images"
_NeurIPS.cc/2025/Conference — NeurIPS 2025 poster_

### Official Review · Reviewer_WTnk · 2025-06-13

**Clarity:** 4
**Significance:** 3
**Originality:** 3
**Rating:** 5
**Confidence:** 5

**Summary:**

This paper addresses a topic of significant theoretical importance and clinical value. Confronting the core challenges of current cancer survival analysis based on whole-slide pathology images (WSIs), namely high computational costs, strong dependency on annotations, and the separation of segmentation and prediction tasks, the authors propose a novel end-to-end framework named ZTSurv. This framework integrates zero-shot tumor microenvironment (TME) segmentation with heterogeneous graph learning, achieving efficient and accurate survival prediction on low-resolution WSIs.

**Questions:**

The method's starting point is a pre-existing TIL map. How would the performance of the zero-shot segmentation and final survival prediction be affected if the initial TIL map is of low quality or contains noise? How robust is the model to this?
(This is just a small personal question, if it is not convenient, the author does not have to specifically add the experimental response, just need to be reflected in future work)

**Ethical Concerns:**

["NO or VERY MINOR ethics concerns only"]

**Final Justification:**

The authors have addressed most of my concerns. I will raise my score to 5.

**Limitations:**

**Weaknesses:**
1. The paper's core contribution includes "zero-shot TME segmentation." According to the paper's definition, this term is used because the model's loss is calculated only on the "lymphocyte" class, with no supervision for "tumor" and "stroma". However, in academic circles, a strict definition of "Zero-shot Learning" typically implies that the model has not seen any samples of the target classes during training. The training WSIs contain all three tissue types simultaneously. The model has visually "seen" tumor and stroma, even if it didn't receive label signals for them. The model receives explicit prior knowledge about the target classes through the LLM-generated text descriptions (e.g., "This is the tumor in the pathology image, typically appearing as..."). This setup is closer to "Zero-label Learning" or a specific form of "Weakly Supervised Learning." While this does not diminish the method's novelty, using the term "Zero-shot" may be debatable to some reviewers. Suggestion: The authors could either more precisely define their use of "zero-shot" in the context of their specific task or add a brief discussion clarifying the similarities and differences with traditional zero-shot learning to avoid ambiguity and make the argument more robust.
2. In the NeurIPS checklist in the appendix (Question 16, "Declaration of LLM usage"), the authors' answer is "[NA]," with the justification that "The core method development in our study does not involve LLMs as any important, original, or non-standard components.". However, in the main body of the paper (Sec 3.1, Sec 4.2) and Appendix D, the authors explicitly state that they used GPT-4 as the LLM to generate detailed text descriptions for different tissue categories. These text descriptions are a crucial part of generating the "prototypes" that guide the zero-shot segmentation and are undoubtedly an important component of the core methodology. These two statements are in direct contradiction.
3. The paper effectively proves that the components of the heterogeneous graph are beneficial for performance through ablation studies. However, it lacks a deeper insight into why they are effective. The authors propose an excellent hypothesis in the introduction: "we have more chance to see the TME component of lymphocyte near the tumor region for long survival patients". Yet, there is no experiment designed to verify if ZTSurv actually learns this pattern.

**Paper Formatting Concerns:**

NO Paper Formatting Concerns

**Quality:**

3

**Strengths And Weaknesses:**

**Strengths:**
1. This paper performs analysis directly on 50x down-sampled low-resolution WSIs, dramatically reducing memory and time costs. This is a massive breakthrough for the scalability and practicality of large-scale clinical data analysis.
2. One of the paper's main highlights is its zero-shot TME segmentation strategy. By using only existing tumor-infiltrating lymphocyte maps as supervision and leveraging a pathology-language foundation model with class descriptions generated by a large language model, it achieves unsupervised segmentation of tumor and stroma regions.
3. When constructing the heterogeneous graph, the authors not only used visual features but also innovatively incorporated semantic features to represent the nodes of different TME components. This dual "visual + semantic" representation enhances the model's ability to distinguish between TME components and captures their complex interactions through edges with continuous attributes, making the graph construction more refined and biologically meaningful.
4. The paper was validated on four major cancer types from TCGA (BRCA, UCEC, LUAD, BLCA), which is representative. It used industry-recognized evaluation metrics such as the C-index, Kaplan-Meier survival curves, and mIoU, ensuring the reliability of the results.

---

> ### Author Rebuttal · Authors · 2025-07-31
>
> Thanks for your constructive comment. Here are the responses to Weaknesses (W) and Questions (Q).
>
> **W1. Definition of zero-shot learning.**
>
> We apologize for the misunderstanding caused. In fact, our method follows a transductive zero-shot learning approach. As a matter of fact, the generalized zero-shot learning can be divided into two settings: inductive and transductive. In the inductive setting, only information from seen classes is used during training. In contrast, the transductive setting allows access to partial information (e.g., class names or visual features) from unseen classes except for the ground truth labels in the training process. We will clarify this distinction in the revised version. Thank you for your valuable suggestion.
>
> **W2. Declaration of LLM usage.**
>
> We apologize for the inconsistency between our response in the NeurIPS checklist and the main body of the paper. We recognize that the use of GPT-4 to generate detailed text descriptions for different tissue categories is crucial in our method. We will update the checklist in the revised version to accurately reflect the role of LLMs in our approach. Thank you for pointing this out.
>
> **W3. There is no experiment designed to verify if ZTSurv actually learns the pattern.**
>
> We thank the reviewer for such insightful comments. Due to the limitation of uploading the visualization results in the rebuttal phase, we instead follow the study in [1], which calculates the mean edge weights between tumor and lymphocyte nodes in the constructed graph for long-survival and short-survival patients based on the patient stratification results in Fig.2. Intuitively, the weights of the edges that connect tumor and lymphocyte should be higher for patients in low survival risk group, since the strong interactions between lymphocyte and tumor regions will indicate a brisk immune response to human cancers [2], and the experimental results denoted below table is also consistent with the above immune knowledge that can demonstrate the effectiveness of our method.
>
> | Patients       | BRCA | UCEC | LUAD | BLCA |
> | -------------- | ---- | ---- | ---- | ---- |
> | Long Survival  | 0.77 | 0.69 | 0.70 | 0.72 |
> | Short Survival | 0.35 | 0.38 | 0.23 | 0.29 |
>
> [1] Zuo, Yingli, et al. "Identify consistent imaging genomic biomarkers for characterizing the survival-associated interactions between tumor-infiltrating lymphocytes and tumors." International Conference on Medical Image Computing and Computer-Assisted Intervention. Cham: Springer Nature Switzerland, 2022.
>
> [2] Saltz, Joel, et al. "Spatial organization and molecular correlation of tumor-infiltrating lymphocytes using deep learning on pathology images." Cell reports 23.1 (2018): 181-193.
>
> **Q1. How would the performance be affected if the initial TIL map is of low quality or contains noise.**
>
> Thanks for this great comment. To assess the robustness of our model to label noise in TIL maps, we randomly perturb 5% and 10% of the labels in the original TIL map and evaluate their performance for both segmentation and survival prediction tasks.
>
> Segmentation results under different label noise ratio:
>
> | Noise Level | BRCA              | UCEC              | LUAD              | BLCA              | Overall |
> | ----------- | ----------------- | ----------------- | ----------------- | ----------------- | ------- |
> | 0%          | 0.574 $\pm$ 0.023 | 0.506 $\pm$ 0.147 | 0.595 $\pm$ 0.023 | 0.608 $\pm$ 0.055 | 0.571   |
> | 5%          | 0.570 $\pm$ 0.027 | 0.494 $\pm$ 0.138 | 0.593 $\pm$ 0.025 | 0.602 $\pm$ 0.054 | 0.565   |
> | 10%         | 0.561 $\pm$ 0.032 | 0.485 $\pm$ 0.142 | 0.585 $\pm$ 0.029 | 0.595 $\pm$ 0.058 | 0.557   |
>
> Survival prediction results under different label noise ratio:
>
> | Noise Level | BRCA              | UCEC              | LUAD              | BLCA              | Overall |
> | ----------- | ----------------- | ----------------- | ----------------- | ----------------- | ------- |
> | 0%          | 0.642 $\pm$ 0.029 | 0.726 $\pm$ 0.113 | 0.637 $\pm$ 0.033 | 0.637 $\pm$ 0.042 | 0.661   |
> | 5%          | 0.639 $\pm$ 0.031 | 0.721 $\pm$ 0.108 | 0.633 $\pm$ 0.040 | 0.636 $\pm$ 0.049 | 0.657   |
> | 10%         | 0.631 $\pm$ 0.035 | 0.714 $\pm$ 0.105 | 0.625 $\pm$ 0.046 | 0.627 $\pm$ 0.051 | 0.649   |
>
> As can be seen from the tables above, although the prediction results for both segmentation and survival prediction tasks are slightly decreased when the label noises are introduced, our method is general robust to the TIL maps with low-level label noise, showing its potential in real-world clinical applications, where perfectly labeled data is often limited.
>
>
>
> Once again, we sincerely thank you for your thoughtful review and constructive feedback. If you have any further questions, we would be glad to respond. If you find our responses satisfactory, we would greatly appreciate your reconsideration of the score.

---

> > ### Comment · Reviewer_WTnk · 2025-08-06
> > **Response to authors**
> >
> > Thank you for your detailed rebuttal, which has addressed most of my concerns. I will raise my score to 5 in response.

---

> > > ### Author Response · Authors · 2025-08-07
> > >
> > > Thank you for taking the time to review our responses. We sincerely appreciate your thoughtful feedback and your engagement throughout the review process. Please feel free to reach me out if you have any other concerns.

---

> ### Author Response · Authors · 2025-08-06
>
> We sincerely appreciate the time and effort you have devoted to reviewing our submission. We have carefully prepared our rebuttal, aiming to address the valuable points you raised.  Should you have an opportunity to review our response, we would be grateful for any additional thoughts or clarifications you may wish to share. Thank you once again for your thoughtful and constructive feedback.
>
> Best regards,
>
> The Authors

---

### Official Review · Reviewer_FFJs · 2025-07-01

**Clarity:** 3
**Significance:** 3
**Originality:** 3
**Rating:** 4
**Confidence:** 3

**Summary:**

This paper proposes ZTSurv, an end-to-end framework for cancer survival prediction using zero-shot segmentation of Tumor Microenvironment (TME) components on low-resolution Whole Slide Images (WSIs). The method leverages Tumor Infiltrating Lymphocyte (TIL) maps to enable annotation-free segmentation of tumor and stroma via a pathology-language foundation model (PLIP). A heterogeneous graph incorporating visual and semantic features then models spatial interactions for survival prediction. Experiments on four TCGA cancer cohorts show superior performance and reduced computational costs compared to state-of-the-art methods.

**Questions:**

1. ZTSurv does not leverage GPT-4-generated class descriptions for node representation—relying instead on basic class names. Why not reuse class descriptions(used in Zero-shot Tumor Microenvironment Segmentation stage) to unify semantic alignment across modules?
2. The paper allows the survival prediction loss to guide the TME segmentation training in reverse (Formula 12: Total loss = Segmentation loss + $\gamma$ × Survival loss), which may lead to:

(1)	The model could mislabel tissues (e.g., classify tumor regions as stroma) if doing so improves survival prediction scores.

(2)	The paper doesn’t check if survival guidance introduces segmentation errors. Survival loss backpropagation to segmentation lacks verification.

3. How are the weight coefficients $\alpha$, $\beta$ and $\gamma$ in the loss function determined?

**Ethical Concerns:**

["NO or VERY MINOR ethics concerns only"]

**Final Justification:**

4: Borderline accept

**Limitations:**

yes

**Quality:**

3

**Strengths And Weaknesses:**

Strengths:

1.	Reduces annotation burden by enabling zero-shot TME segmentation without pixel-level pathologist annotation.

2.	Exceptional efficiency with 3,000× lower memory and 6× faster inference via 50× down-sampled WSIs compared to patch-based methods.

3.	Extensive validation across four cancer types with multiple metrics (C-index, mIoU, Kaplan-Meier) demonstrates robustness.

Weaknesses:
1. ZTSurv enables zero-shot segmentation on other two important TME components (i.e., tumor and stroma) but relies on TIL maps requiring manual labeling, transferring annotation burden upstream.
2. While the paper proposes zero-shot segmentation for tumor and stroma, they only validates lymphocyte predictions via mIoU in Table 2, lacking ground-truth validation of tumor and stroma and influencing clinical trustworthiness.
3. ZTSurv's tissue ablation study in Table 3 excludes necrosis—a validated independent survival predictor. This contrasts with ProtoSurv's[1] inclusion of necrosis in its 5-class framework in Table 4, where comprehensive ablation proved necrosis essential for optimal prognosis modeling.
4. Graph construction based on KNN only relies on feature similarity and ignores spatial distance. Verify whether the current method is superior to the traditional spatial graph methods.
5. The evaluation of mIoU is only performed on lymphocytes with ground truth. For tumor and stroma, there are only a few selected visualization results (Figure 3), which are far from convincing.

[1] Leveraging Tumor Heterogeneity: Heterogeneous Graph Representation Learning for Cancer Survival Prediction in Whole Slide Images

---

> ### Author Rebuttal · Authors · 2025-07-31
>
> Thanks for your constructive comment. Here are the responses to Weaknesses (W) and Questions (Q).
>
> **W1. ZTSurv transfers annotation burden upstream.**
>
> Previous methods require training a segmentation or classification model that includes all target classes, necessitating manual annotations for each of these categories. In contrast, our approach only requires annotation of the TIL map, significantly reducing the annotation burden.
>
> **W2 & W5. Zero-shot segmentation lacks ground-truth validation of tumor and stroma.**
>
> We thank the reviewer for such great comment. We would like to clarify that currently, there is no complete semantic segmentation dataset available for TCGA, and the only existing annotation is the TIL map. As a result, quantitative evaluation is limited to the seen class (lymphocyte), for which we report mIoU. Since ground truth annotations for unseen classes (tumor and stroma) are unavailable, direct evaluation of these classes is not feasible. To address this limitation and assess the reliability of our segmentation results, we leverage the foundation model PLIP to compute the image–text alignment between segmented image patches and class-specific text prompts. Specifically, we evaluate the similarity between: (1) tumor patches and the text prompt “tumor”, (2) tumor patches and the text prompt “stroma”, (3) stroma patches and the text prompt“tumor”, and (4) stroma patches and the text prompt“stroma”. We compute the average scores for each of these pairs across four datasets. The results are presented in the table below.
>
> | Patch-Text                   | BRCA | UCEC | LUAD | BLCA |
> | ---------------------------- | ---- | ---- | ---- | ---- |
> | Tumor (Image)-Tumor (Text)   | 0.85 | 0.88 | 0.83 | 0.86 |
> | Tumor( Image)-Stroma (Text)  | 0.31 | 0.33 | 0.29 | 0.35 |
> | Stroma (Image)-Tumor (Text)  | 0.28 | 0.30 | 0.27 | 0.33 |
> | Stroma (Image)-Stroma (Text) | 0.82 | 0.80 | 0.79 | 0.81 |
>
> From the results, we observe that the similarity between image and text embeddings of the same tissue type is consistently higher than that between different tissue types, which to some extent reflects the accuracy of our segmentation for unseen classes (tumor, stroma).
>
> **W3. ZTSurv's tissue ablation study in Table 3 excludes necrosis.**
>
> We thank the reviewer for such great comment. We conduct an additional experiment incorporating necrosis, and here are the results:
>
> |                                        | BRCA              | UCEC              | LUAD              | BLCA              | Overall |
> | -------------------------------------- | ----------------- | ----------------- | ----------------- | ----------------- | ------- |
> | lymphocyte + tumor + stroma            | 0.642 $\pm$ 0.029 | 0.726 $\pm$ 0.113 | 0.637 $\pm$ 0.033 | 0.637 $\pm$ 0.042 | 0.661   |
> | lymphocyte + tumor + stroma + necrosis | 0.651 $\pm$ 0.031 | 0.739 $\pm$ 0.117 | 0.647 $\pm$ 0.025 | 0.649 $\pm$ 0.045 | 0.672   |
>
> From the experimental results, it is evident that including necrosis indeed improves survival prediction, which aligns with the findings in [1]. We acknowledge this limitation in the Appendix H.7, where we note that including a broader range of TME components, such as blood vessels, necrosis, and fibroblasts, could further enhance predictive performance. In future work, we will explore these additional tissue types to capture a more comprehensive representation of the TME and improve the robustness of our model.
>
> [1] Wu, Junxian, et al. "Leveraging tumor heterogeneity: Heterogeneous graph representation learning for cancer survival prediction in whole slide images." Advances in Neural Information Processing Systems 37 (2024): 64312-64337.
>
> **W4. Graph construction based on KNN only relies on feature similarity and ignores spatial distance.**
>
> We apologize for the confusion. To clarify, our graph construction method based on KNN incorporates spatial distance (L2 norm). We will correct this in the revised version. Furthermore, to address this concern more directly, we conduct an additional experiment where the graph is constructed based on feature similarity. Here are the results:
>
> |                    | BRCA              | UCEC              | LUAD              | BLCA              | Overall |
> | ------------------ | ----------------- | ----------------- | ----------------- | ----------------- | ------- |
> | feature similarity | 0.639 $\pm$ 0.031 | 0.726 $\pm$ 0.097 | 0.635 $\pm$ 0.028 | 0.638 $\pm$ 0.036 | 0.660   |
> | spatial distance   | 0.642 $\pm$ 0.029 | 0.726 $\pm$ 0.113 | 0.637 $\pm$ 0.033 | 0.637 $\pm$ 0.042 | 0.661   |
>
> **Q1. ZTSurv does not leverage GPT-4-generated class descriptions for node representation.**
>
> We appreciate this valuable suggestion. We conduct a comparison experiment where we incorporate GPT-4-generated class descriptions for node representation. Here are the results:
>
> | Node Representation | BRCA              | UCEC              | LUAD              | BLCA              | Overall |
> | ------------------- | ----------------- | ----------------- | ----------------- | ----------------- | ------- |
> | class name          | 0.642 $\pm$ 0.029 | 0.726 $\pm$ 0.113 | 0.637 $\pm$ 0.033 | 0.637 $\pm$ 0.042 | 0.661   |
> | class description   | 0.651 $\pm$ 0.026 | 0.737 $\pm$ 0.120 | 0.644 $\pm$ 0.028 | 0.645 $\pm$ 0.039 | 0.669   |
>
> From the results, incorporating GPT-4-generated class descriptions for node representation contributes to improved performance on multiple datasets. We again appreciate your insightful suggestion.
>
> **Q2. Survival loss backpropagation to segmentation lacks verification.  & Q3. How are the weight coefficients in the loss function determined.**
>
> We thank the reviewer for raising these important questions. Recent studies have demonstrated that accurate modeling of the TME is critical for reliable survival prediction [1] [2] [3]. This suggests that segmentation and survival prediction are inherently aligned objectives in the context of histopathology analysis. To verify the effect of survival-guided segmentation, we compare the performance of zero-shot segmentation with and without survival guidance in Table 2 of the original manuscript. The results indicate that incorporating survival supervision leads to improved segmentation accuracy, supporting the effectiveness of this joint optimization strategy.
>
> However, we agree with the reviewer that the balance between segmentation and survival loss is crucial. In our formulation, this trade-off is controlled by the hyperparameter $\gamma$. To examine its effect, we conduct experiments under different $\gamma$ values and report both survival prediction and segmentation performance across four datasets.
>
> Segmentation results under different $\gamma$ values:
>
> | $\gamma$ | BRCA              | UCEC              | LUAD              | BLCA              | Overall |
> | -------- | ----------------- | ----------------- | ----------------- | ----------------- | ------- |
> | 5        | 0.556 $\pm$ 0.025 | 0.482 $\pm$ 0.034 | 0.581 $\pm$ 0.019 | 0.590 $\pm$ 0.026 | 0.552   |
> | 10       | 0.561 $\pm$ 0.028 | 0.500 $\pm$ 0.131 | 0.589 $\pm$ 0.021 | 0.595 $\pm$ 0.028 | 0.561   |
> | 20       | 0.574 $\pm$ 0.023 | 0.506 $\pm$ 0.147 | 0.595 $\pm$ 0.023 | 0.608 $\pm$ 0.055 | 0.571   |
> | 50       | 0.546 $\pm$ 0.026 | 0.475 $\pm$ 0.116 | 0.570 $\pm$ 0.020 | 0.582 $\pm$ 0.049 | 0.543   |
>
> Survival prediction results under different $\gamma$ values:
>
> | $\gamma$ | BRCA              | UCEC              | LUAD              | BLCA              | Overall |
> | -------- | ----------------- | ----------------- | ----------------- | ----------------- | ------- |
> | 5        | 0.635 $\pm$ 0.030 | 0.713 $\pm$ 0.118 | 0.624 $\pm$ 0.036 | 0.629 $\pm$ 0.040 | 0.650   |
> | 10       | 0.641 $\pm$ 0.028 | 0.723 $\pm$ 0.112 | 0.632 $\pm$ 0.039 | 0.635 $\pm$ 0.038 | 0.658   |
> | 20       | 0.642 $\pm$ 0.029 | 0.726 $\pm$ 0.113 | 0.637 $\pm$ 0.033 | 0.637 $\pm$ 0.042 | 0.661   |
> | 50       | 0.631 $\pm$ 0.031 | 0.710 $\pm$ 0.114 | 0.624 $\pm$ 0.037 | 0.627 $\pm$ 0.041 | 0.648   |
>
> From these results, we observe that when $\gamma$ is set to a very small value or to 0, as in the original Table 2 where no survival guidance is applied, segmentation lacks prognostic supervision and tends to capture less clinically meaningful structures, leading to reduced segmentation accuracy. In contrast, when $\gamma$ is too large (for example, $\gamma$=50), the survival loss dominates the training objective. This can mislead the segmentation module, as it begins to overfit to survival-related global signals at the expense of precise spatial delineation.
>
> For the other coefficients in the loss function i.e., $\alpha$ and $\beta$ are set following the configurations from previous work [4].
>
> [1] Wu, Junxian, et al. "Leveraging tumor heterogeneity: Heterogeneous graph representation learning for cancer survival prediction in whole slide images." Advances in Neural Information Processing Systems 37 (2024): 64312-64337.
>
> [2] Shao, Wei, et al. "Tumor micro-environment interactions guided graph learning for survival analysis of human cancers from whole-slide pathological images." Proceedings of the IEEE/CVF Conference on Computer Vision and Pattern Recognition. 2024.
>
> [3] Han, Minghao, et al. "Multi-scale heterogeneity-aware hypergraph representation for histopathology whole slide images." 2024 IEEE International Conference on Multimedia and Expo (ICME). IEEE, 2024.
>
> [4] Zhou, Ziqin, et al. "Zegclip: Towards adapting clip for zero-shot semantic segmentation." Proceedings of the IEEE/CVF conference on computer vision and pattern recognition. 2023.
>
>
>
> Once again, we sincerely thank you for your thoughtful review and constructive feedback. If you have any further questions, we would be glad to respond. If you find our responses satisfactory, we would greatly appreciate your reconsideration of the score.

---

> > ### Comment · Reviewer_FFJs · 2025-08-06
> >
> > Thanks for your responses. I will maintain my scores.

---

> > > ### Author Response · Authors · 2025-08-06
> > >
> > > Thank you for taking the time to review our responses. We sincerely appreciate your thoughtful feedback and your engagement throughout the review process. Please feel free to reach me out if you have any other concerns.

---

> ### Author Response · Authors · 2025-08-06
>
> We sincerely appreciate the time and effort you have devoted to reviewing our submission. We have carefully prepared our rebuttal, aiming to address the valuable points you raised.  Should you have an opportunity to review our response, we would be grateful for any additional thoughts or clarifications you may wish to share. Thank you once again for your thoughtful and constructive feedback.
>
> Best regards,
>
> The Authors

---

### Official Review · Reviewer_Ectr · 2025-07-02

**Clarity:** 3
**Significance:** 3
**Originality:** 3
**Rating:** 4
**Confidence:** 4

**Summary:**

The authors propose a method to address four key challenges in computational pathology for survival prediction: (1) the high computational cost of patch-based processing for gigapixel Whole-Slide Images (WSIs), (2) the limited capacity of existing methods to comprehensively model the Tumor Micro-Environment (TME) and its spatial organization, (3) the reliance of current GNN-based approaches on a two-step pipeline involving pre-segmented TME components, and (4) the inability of standard GNNs to jointly incorporate visual features and semantic information in node representations. To overcome these limitations, the authors introduce ZTSurv, a framework for cancer survival prediction that simultaneously segments TME components, starting from TIL maps, by leveraging pathology-language foundation models. These models fuse semantic and visual information to build heterogeneous graphs over downsampled (50x) WSIs, reducing computational demands. The method is evaluated across four TCGA cancer cohorts, with supporting ablation studies and survival stratification analyses.

**Questions:**

1. How does the proposed method compare to recent slide-level foundation models (e.g., TITAN, PRISM) that are explicitly designed to capture global spatial and contextual relationships across whole-slide images for survival prediction? Why are these models not considered in the comparisons or discussed in the related work?
2. Given that the model operates on 50x downsampled WSIs, how does this resolution affect the segmentation quality of fine-grained structures such as TILs and stromal boundaries, and how does it influence the survival prediction performance? Have the authors conducted any ablation studies or resolution comparisons to quantify this trade-off?
3. Can the authors justify the choice of labeling graph patches using the dominant tissue class within fixed-size regions, especially in areas with high tissue heterogeneity? How do they account for the potential label noise introduced at tissue boundaries or mixed-type regions?
4. The graph edge construction relies on Pearson correlation between concatenated visual and semantic embeddings. How do the authors ensure that this metric produces biologically meaningful relationships, given the statistical and functional differences between visual and text features? Have alternative or learned similarity measures been considered?
5. Can the authors clarify how the baseline models were trained, particularly in terms of input resolution? Were they evaluated using full-resolution WSIs or similarly downsampled inputs (e.g., 50x), and how does this affect the fairness and validity of the performance comparison?

**Ethical Concerns:**

["NO or VERY MINOR ethics concerns only"]

**Final Justification:**

After reading the authors’ responses, I find that three of the five main weaknesses I previously identified have been satisfactorily addressed: (W1-2-5) are now supported by new experiments.

Two points of improvement remain:

- (W3) While the authors acknowledge the limitation raised and plan to include it in the revised manuscript, the concern remains valid. Assigning a dominant tissue label to each graph patch, though consistent with prior works [1][2], may be suboptimal in the presence of heterogeneous tissue regions. The suggested trade-off between patch size and computational complexity is noted, but the issue underscores a limitation in the method’s ability to capture fine-grained heterogeneity. This remains a relevant weakness for the current version of the work.
- (W4) The authors provide an extensive response supported by new experiments to the concern regarding the use of Pearson correlation on concatenated image-text embeddings, and the additional similarity analysis is appreciated. However, while the supplemental learned similarity experiment shows a slight improvement over the fixed Pearson metric, the performance gains are relatively marginal. The authors acknowledge that Pearson correlation may not fully capture non-linear interactions, yet the proposed learnable attention-based metric will be only briefly explored in the current version of the manuscript. The reliance on a fixed similarity function in the main method still constitutes a limitation, particularly given the known complexity of interactions in the tumor microenvironment. Further investigation, that the authors remand to future work, into more expressive or adaptive similarity functions would strengthen the methodology.

I have also read all other reviews and the authors’ responses. Considering that most concerns have been addressed, aside from the two remaining weaknesses highlighted, I would maintain my score as 4 (Borderline accept)

**Limitations:**

The authors adequately addressed the limitations and potential negative societal impact of their work

**Paper Formatting Concerns:**

No Paper Formatting Concerns

**Quality:**

2

**Strengths And Weaknesses:**

Strengths
1. The manuscript is clearly written and well-organized. While leveraging segmented regions of interest to enhance task performance is a common approach in the literature, the proposed framework introduces a novel and effective perspective by enabling survival prediction and tissue segmentation to co-evolve and mutually reinforce each other during training. Moreover, the model’s ability to discover new tissue compartments, such as stroma and cancerous regions, without requiring pixel-level supervision holds promising implications for broader applications.

Weaknesses
1. While the manuscript "critiques" patch-based aggregation strategies for their inability to capture the spatial heterogeneity of the tumor microenvironment, it does not acknowledge recent advances in slide-level foundation models, such as TITAN [1], PRISM [2], CHIEF[3]…, that are explicitly designed to overcome these limitations by modeling global spatial and contextual relationships across the whole slide. This omission weakens the positioning of the proposed method with respect to the current state-of-the-art.
[1] Ding, Tong, et al. "Multimodal whole slide foundation model for pathology." arXiv preprint arXiv:2411.19666 (2024).
[2] Shaikovski, George, et al. "Prism: A multi-modal generative foundation model for slide-level histopathology." arXiv preprint arXiv:2405.10254 (2024).
[3] Wang, Xiyue, et al. "A pathology foundation model for cancer diagnosis and prognosis prediction." Nature 634.8035 (2024): 970-978.
2. Even though the proposed method outperforms state-of-the-art approaches (which are presumably trained at original WSI resolution), the authors report reduced computational cost by operating on 50x downsampled WSIs, this design choice raises concerns about the loss of fine-grained information, particularly for tasks such as TILs or stromal boundary segmentation and survival analysis, which rely on high-resolution (0.25-0.5mpp) histological detail. The manuscript lacks an ablation or comparative study to assess the impact of resolution on segmentation accuracy and survival prediction performance. Moreover, no quantitative analysis of runtime or efficiency is provided to justify the proposed trade-off.
3. The graph is constructed by dividing downsampled WSIs into fixed-size patches and assigning a tissue label based on the dominant class within each patch. While computationally efficient, this heuristic may oversimplify the true spatial heterogeneity of the tumor microenvironment. In regions where multiple tissue types (e.g., tumor-stroma or tumor-lymphocyte interfaces) are closely mixed, assigning a single label to the entire patch risks introducing significant label noise. This can lead to ambiguous or misleading node representations, particularly at boundaries where biologically meaningful transitions occur. Such simplification may undermine the expressiveness of the graph, especially given that downstream survival prediction depends on accurate modeling of fine-grained TME interactions. Without explicit modeling of sub-patch level composition or uncertainty, the proposed node construction method may fail to capture critical spatial cues that are central to both segmentation fidelity and survival risk modeling.
4. The edge definitions in the graph are based on Pearson correlation computed over concatenated visual and text embeddings from the PLIP encoder. While this fusion aims to create a unified node representation, it merges two fundamentally distinct modalities, visual features (e.g., morphology, texture) and semantic cues (e.g., tissue type, biological role), which differ in both statistical properties and functional relevance. Pearson correlation assumes a linear relationship and equal scaling across dimensions, an assumption that may not hold in this multimodal context. As a result, nodes that are semantically dissimilar but visually similar (or vice versa) may be incorrectly assigned strong connections, potentially introducing biologically implausible edges. Although the authors apply random edge and feature dropout as a generic regularization mechanism to mitigate noisy or spurious correlations, this does not address the underlying limitation of using a fixed, hand-crafted similarity metric that lacks biological interpretability. A more principled solution would involve decoupling the visual and semantic similarity pathways or learning edge weights in a data-driven manner, e.g., through attention-based mechanisms, to better reflect meaningful relationships in the tumor microenvironment.

Minor Weakness: typo at line 283 “furthrt”

---

> ### Author Rebuttal · Authors · 2025-07-31
>
> Thanks for your constructive comment. Here are the responses to Weaknesses (W) and Questions (Q).
>
> **W1&Q1. How does the proposed method compare to recent slide-level foundation models.**
>
> Thanks for this great comment. We agree that recent slide-level foundation models such as TITAN, PRISM, CHIEF are promising advances toward modeling spatial and contextual information across WSIs. However, they typically adopt a two-stage design, where the modeling of TME structures and survival prediction are handled independently that miss the inherent correlation among them. Moreover, they rely heavily on high-resolution patch-based preprocessing, which requires considerable computation and memory for tiling and patch-level feature storage. In contrast, our method simultaneously realize the TME segmentation and survival prediction tasks in an end-to-end manner on the down-sampled WSIs, which enables joint modeling of spatial tissue structures and prognostic outcomes while reducing computational cost.
>
> We conduct a comparison using the WSI-level embeddings generated by TITAN, PRISM, and CHIEF for survival prediction. As shown below, our approach outperforms these methods in survival prediction across four cancer types. Moreover, according to your suggestions, we will introduce these slide-level foundation models in the related work of the revised manuscript.
>
> |Model|BRCA|UCEC|LUAD|BLCA|Overall|
> |-|-|-|-|-|-|
> |TITAN|0.614±0.025|0.697±0.031|0.613±0.017|0.610±0.028|0.634|
> |PRISM|0.607±0.022|0.690±0.028|0.607±0.023|0.605±0.037|0.627|
> |CHIEF|0.632±0.027|0.694±0.034|0.641±0.029|0.628±0.031|0.641|
> | **ZTSurv (ours)** | **0.642±0.029** | **0.726±0.113**| **0.637±0.033**| **0.637±0.042**| **0.661**|
>
> **W2&Q2. How does the resolution affect the segmentation and survival prediction performance.**
>
> We fully agree that spatial resolution can influence performance, particularly for tasks that rely on fine-grained histological detail. However, our method is specifically designed to capture the distributional patterns of TME components such as TILs at the whole-slide level, rather than relying on detailed cellular morphology. As demonstrated in [1], the spatial patterns of TILs are closely associated with patients’ survival, highlighting the power of spatial distribution analysis in addressing clinically relevant questions, even without high-resolution morphological detail.
>
> Moreover, as discussed in the response to W3 of reviewer UaL5, we chose 50x downsampling because the utilized TIL maps are labeled under such resolution [1], and thus we are unable to test the performance of our method on the TIL maps with higher resolution. By contrast, according to your suggestion, we conduct additional experiments to examine the impact of lower-resolution inputs by further downsampling the TIL map to 100×. The results are shown below, along with the average inference time per WSI implemented on a single NVIDIA RTX 4090 GPU.
>
> Survival prediction results:
>
> |Downsampling|Avg. Inference Time per WSI (s)|BRCA|UCEC|LUAD|BLCA|Overall|
> |-|-|-|-|-|-|-|
> |50x|32|0.642±0.029|0.726±0.113|0.637±0.033|0.637±0.042|0.661|
> |100x|15|0.602±0.027|0.682±0.030|0.601±0.028|0.603±0.037|0.622|
>
> Segmentation results:
>
> |Downsampling|BRCA|UCEC|LUAD|BLCA|Overall|
> |-|-|-|-|-|-|
> |50x|0.574±0.023|0.506±0.147|0.595±0.023|0.608±0.055|0.571|
> |100x|0.552±0.024|0.472±0.129|0.562±0.026|0.573±0.048|0.540|
>
> As shown in above tables, lowering the resolution to 100× leads to a consistent decline in performance across all cohorts. This suggests that excessive downsampling reduces the amount of useful information available for both tissue segmentation and survival prediction.
>
> [1] Saltz, Joel, et al. "Spatial organization and molecular correlation of tumor-infiltrating lymphocytes using deep learning on pathology images." Cell Reports 23.1 (2018): 181-193.e7.
>
> **W3. Potential label noise introduced at tissue boundaries or mixed-type regions.**
>
> We thank the reviewer for such insightful comment. Actually, the original WSIs are typically scanned at a spatial resolution of 0.25 μm/pixel. After downsampling the images by a factor of 50, the resulting spatial resolution becomes: 0.25 μm/pixel × 50 = 12.5 μm/pixel. This resolution aligns well with the average size of a single cell, which is typically around 10–15 μm in diameter.
>
> At this scale, each pixel approximately corresponds to the size of a cell with specific type, and thus the scenario where each pixel in the down-sampled WSIs corresponds to a different tissue type is unlikely happen. By contrast, if the downsampling level achieves to 100x (i.e., 25.0 μm/pixel), each pixel in the down-sampled image will have more chance to involve multiple cells with diverse types that will lead to ambiguous or misleading node representations. We believe this is the one of the reasons that the survival prediction results under the downsampling ratio of 50x is significantly better to that under 100x. Please see the result tables shown in the answer to your previous comment (W2 &Q2).
>
> **Q3. Dominant tissue class within fixed-size regions.**
>
> We thank the reviewer for such great comment. Similar to previous works [1][2], we assign each graph patch a dominant tissue class label based on the most frequent tissue type within the patch, which has been shown effective in prior studies and our methods. We agree with you that such approximation maybe ineffective on tissues with high heterogeneity. A possible solution is to treat the patch with reduced size as node, but it will bring more graph nodes that will lead to higher computational complexity for graph learning. We will add it as the limitation of this study in the revised manuscript.
>
> [1] Wu, Junxian, et al. "Leveraging tumor heterogeneity: Heterogeneous graph representation learning for cancer survival prediction in whole slide images." Advances in Neural Information Processing Systems 37 (2024): 64312-64337.
>
> [2] Shao, Wei, et al. "Tumor micro-environment interactions guided graph learning for survival analysis of human cancers from whole-slide pathological images." Proceedings of the IEEE/CVF Conference on Computer Vision and Pattern Recognition. 2024.
>
> **W4&Q4. Similarity measures for graph edge construction.**
>
> We appreciate the reviewer’s thoughtful comments. To investigate the concern that Pearson correlation over concatenated visual and text embeddings may create spurious edges, we conduct a controlled similarity analysis across different types of node pair. Specifically, we compute the mean similarity scores for three types of relationships: (1) Image–Image (Same Type): Pairwise similarity between nodes of the same tissue class based on visual features only. (2) Image–Text (Same Type): Similarity between a node's image embedding and its corresponding text embedding (e.g., tumor image vs. “tumor” text). (3) Image–Text (Different Type): Similarity between a node's image embedding and text embeddings from different tissue classes (e.g., tumor image vs. “stroma” text). The results are summarized below:
>
> |Type|Mean Similarity|Std Dev|
> |-|-|-|
> |Image–Image (Same Type)|0.698|0.018|
> |Image–Text (Same Type)|0.341|0.028|
> |Image–Text (Different Type)|0.103|0.043|
>
> From the results, we can observe that image and text embeddings of specific tissues are biologically related but not excessively redundant , which suggests that these two modalities provide complementary information. Consequently, computing Pearson correlation over their concatenated representations offers a meaningful and interpretable way to model interactions within the tumor microenvironment.
>
> We also agree with the reviewer that a fixed similarity metric such as Pearson correlation might not fully capture complex or non-linear relationships. To explore this further, we implement a learnable similarity function inspired by attention mechanisms. Specifically, we project node embeddings into query and key spaces using learnable matrices and compute edge weights by taking the dot product between queries and keys, followed by a ReLU activation. Here are the results:
>
> |Method|BRCA|UCEC|LUAD|BLCA|Overall|
> |-|-|-|-|-|-|
> |Learned similarity|0.647±0.030|0.734±0.117|0.642±0.035|0.644±0.038|0.667|
> |Pearson similarity|0.642±0.029|0.726±0.113|0.637±0.033|0.637±0.042|0.661|
>
> As shown in the table, the learned similarity approach achieves improved performance to fixed Pearson correlation. As part of our future work, we plan to conduct more detailed analyses and further explore the potential of data-driven approaches in this context.
>
> **Q5. How the baseline models were trained.**
>
> All baseline models were trained on full-resolution WSIs, following their original implementations. These methods rely on patch-level inputs extracted at high magnification and cannot be directly applied to down-sampled whole-slide representations. As a result, preprocessing steps such as tiling and patch-level feature extraction are required to prepare the data for these methods.
>
> To further evaluate performance under a consistent input setting, we additionally compare our method with ProtoSurv [1] on 50× down-sampled WSIs. The results are shown below:
>
> |   |BRCA|UCEC|LUAD|BLCA|Overall|
> |-|-|-|-|-|-|
> |ProtoSurv|0.596±0.035|0.559±0.119|0.563±0.048|0.572±0.038|0.573|
> |ZTSurv (ours)|0.642±0.029|0.726±0.113|0.637±0.033|0.637±0.042|0.661|
>
> These results demonstrate that our method maintains strong performance at significantly lower resolution, highlighting its efficiency and scalability.
>
> [1] Wu, Junxian, et al. "Leveraging tumor heterogeneity: Heterogeneous graph representation learning for cancer survival prediction in whole slide images." Advances in Neural Information Processing Systems 37 (2024): 64312-64337.
>
> Once again, we sincerely thank you for your thoughtful review and constructive feedback. If you have any further questions, we would be glad to respond. If you find our responses satisfactory, we would greatly appreciate your reconsideration of the score.

---

> > ### Comment · Reviewer_Ectr · 2025-08-03
> >
> > Thank you to the authors for the rebuttal. I found the response informative and appreciate the additional experiments (especially for W4&Q4). However, I have a few follow-up questions for clarification.
> >
> > W1&Q1: Thank you for the detailed comparison and clarification. To better contextualize the reported results for TITAN, PRISM, and CHIEF, could you please elaborate on which method was used to perform survival analysis using their WSI-level embeddings? Specifically:
> > - Were the embeddings fed into a Cox proportional hazards model, a deep survival network, or another approach?
> > - Were any fine-tuning or projection layers added on top of the embeddings before survival modeling?
> > - Was the survival prediction pipeline standardized across all models, including your own, to ensure a fair comparison?
> >
> > W2&Q2:
> > Thank you for the additional experiments and clarification. While I agree that the spatial distribution of TILs, rather than fine-grained morphology, is a powerful predictor of survival as shown in [1], your own results indicate that increasing resolution from 100× to 50× leads to notable gains in both segmentation accuracy and survival prediction. This suggests that resolution still plays a significant role, even if high histological detail is not fully exploited.
> > Given this trend, have the authors considered exploring intermediate resolutions between 50× and 100× or even slightly higher than 50×, to better understand the trade-off between performance and inference time? In addition, for what concerns resolutions slightly higher than 50×, would it be feasible to upsample the 50× TIL masks, either via pixel-wise interpolation or vectorized polygon conversion for efficiency in processing, to align with higher resolution WSIs? Such an analysis could help identify a more optimal resolution that balances accuracy and efficiency, especially given the clear runtime difference observed between 50× and 100× inputs.

---

> > > ### Author Response · Authors · 2025-08-05
> > >
> > > Thank you for your thoughtful follow-up and for taking the time to engage with our responses.
> > >
> > > **W1 & Q1. Details of survival analysis comparison.**
> > >
> > > To ensure a fair and consistent comparison, we standardize the survival analysis pipeline across all models. Specifically, the WSI-level embeddings obtained from TITAN, PRISM, CHIEF, and our method are all fine-tuned with the same multi-layer perception layers, followed by survival prediction using a Cox proportional hazards model.
> > >
> > > **W2 & Q2. Discussion of the resolution.**
> > >
> > > According to your suggestion, we conduct additional experiments at resolutions of 75× and 30× to further investigate the trade-off between survival prediction performance versus inference time. For higher resolutions, we upsample the original 50× TIL maps using pixel-wise interpolation to align them with higher-resolution WSIs. Due to time constraints, we perform these experiments on two datasets (BRCA and UCEC) and will provide the remaining results for LUAD and BLCA as soon as they are available.
> > >
> > > | Downsampling | Avg. Inference Time per WSI (s) | BRCA        | UCEC        |
> > > | ------------ | ------------------------------- | ----------- | ----------- |
> > > | 30x          | 56                              | 0.636±0.024 | 0.719±0.101 |
> > > | 50x          | 32                              | 0.642±0.029 | 0.726±0.113 |
> > > | 75x          | 22                              | 0.619±0.029 | 0.704±0.091 |
> > > | 100x         | 15                              | 0.602±0.027 | 0.682±0.030 |
> > >
> > > From the results, we observe a performance drop at lower resolutions (e.g., 75× and 100×), potentially due to the loss of spatial detail critical for accurate TIL localization and survival modeling. At higher resolution (30×), the performance is slightly inferior to that at 50× despite the increased input detail, which may be attributed to the reduced reliability of the interpolated TIL maps, as the interpolation process can introduce boundary artifacts and label noise, compromising supervision quality.

---

> > > ### Author Response · Authors · 2025-08-07
> > >
> > > We thank the reviewer again for the valuable suggestion regarding exploring intermediate and higher resolutions. Following our previous response where we report results on two datasets, we have now completed the additional experiments on the remaining two datasets (LUAD and BLCA). The updated results for all four datasets at multiple downsampling levels are summarized in the tables below.
> > >
> > > | Downsampling | BRCA        | UCEC        | LUAD        | BLCA        | Overall |
> > > | ------------ | ----------- | ----------- | ----------- | ----------- | ------- |
> > > | 30x          | 0.636±0.024 | 0.719±0.101 | 0.633±0.029 | 0.631±0.040 | 0.655   |
> > > | 50x          | 0.642±0.029 | 0.726±0.113 | 0.637±0.033 | 0.637±0.042 | 0.661   |
> > > | 75x          | 0.616±0.029 | 0.704±0.091 | 0.624±0.031 | 0.613±0.032 | 0.639   |
> > > | 100x         | 0.602±0.027 | 0.682±0.030 | 0.601±0.028 | 0.603±0.037 | 0.622   |

---

> > > > ### Comment · Reviewer_Ectr · 2025-08-07
> > > >
> > > > After reading the authors’ responses, I find that three of the five main weaknesses I previously identified have been satisfactorily addressed: (W1-2-5) are now supported by new experiments.
> > > >
> > > > Two points of improvement remain:
> > > > - (W3) While the authors acknowledge the limitation raised and plan to include it in the revised manuscript, the concern remains valid. Assigning a dominant tissue label to each graph patch, though consistent with prior works [1][2], may be suboptimal in the presence of heterogeneous tissue regions. The suggested trade-off between patch size and computational complexity is noted, but the issue underscores a limitation in the method’s ability to capture fine-grained heterogeneity. This remains a relevant weakness for the current version of the work.
> > > > - (W4) The authors provide an extensive response supported by new experiments to the concern regarding the use of Pearson correlation on concatenated image-text embeddings, and the additional similarity analysis is appreciated. However, while the supplemental learned similarity experiment shows a slight improvement over the fixed Pearson metric, the performance gains are relatively marginal. The authors acknowledge that Pearson correlation may not fully capture non-linear interactions, yet the proposed learnable attention-based metric will be only briefly explored in the current version of the manuscript. The reliance on a fixed similarity function in the main method still constitutes a limitation, particularly given the known complexity of interactions in the tumor microenvironment. Further investigation, that the authors remand to future work, into more expressive or adaptive similarity functions would strengthen the methodology
> > > >
> > > > I have also read all other reviews and the authors’ responses. Considering that most concerns have been addressed, aside from the two remaining weaknesses highlighted, I would maintain my score while I will remain engaged during the rebuttal period. I thank the authors again for their work and time.

---

> > > > > ### Author Response · Authors · 2025-08-07
> > > > >
> > > > > We appreciate your thoughtful assessment and recognition that three key concerns (W1, W2, W5) have been addressed through our additional experiments.
> > > > >
> > > > > We would also like to take this opportunity to reiterate the primary contribution of our work. In this work, we introduces a novel end-to-end framework that jointly performs zero-shot tissue segmentation and survival prediction on low-resolution WSIs. To the best of our knowledge, this is the first unified approach that integrates both tasks in a zero-shot setting, which helps reduce annotation burden and improve survival prediction performance, while substantially lowering computational and storage costs compared to existing high-resolution methods.
> > > > >
> > > > > Regarding the remaining points (W3, W4), we recognize these as valuable directions for further improvement. We will clearly note these aspects in the revised manuscript and consider exploring more flexible patch labeling strategies and advanced similarity modeling techniques in future work.
> > > > >
> > > > > Thank you again for your insightful feedback and constructive engagement throughout the review process.

---

### Official Review · Reviewer_UaL5 · 2025-07-05

**Clarity:** 3
**Significance:** 2
**Originality:** 2
**Rating:** 4
**Confidence:** 4

**Summary:**

In pathological whole slide images (WSI), prognosis is significantly associated with tumor micro-environment (TME) components, with lymphocyte, stroma and tumor areas being part of this. Training a model to make use of these is difficult due to high resolution of WSIs as well as the lack of annotation of TME components. Authors propose ZTSurv: A model that makes use of PLIP text embeddings to zero-shot predict stroma and tumor areas in a heavily (50x) downsampled WSI, while being directly supervised only for lymphocyte segmentation. Then, a graph of these different regions is constructed and input to a graph convolutional (GCN) network for survival prediction. Experiments on four cancer cohorts from TCGA illustrates superior C-index compared to competing methods.

**Questions:**

I've summarized my concerns and recommendations in the "Weaknesses" section. I am slightly veering on acceptance due to the strength and applicability of the proposed model. If the rebuttal can answer my concerns, I'd gladly revise my score based on further insights. I thank the authors for their work and time.

**Ethical Concerns:**

["NO or VERY MINOR ethics concerns only"]

**Final Justification:**

Based on the points I raised in my comments to the authors' rebuttal, I keep my initial score of 4, which I find to be still appropriate due to the lack of significant technical contribution with the knowledge that the paper targets a specific application (pathology).

**Limitations:**

Yes, the authors address the limitations of their work in supplementary material.

**Paper Formatting Concerns:**

None that I noted.

**Quality:**

3

**Strengths And Weaknesses:**

Strengths

* Superior performance compared to competing methods makes this approach useful for survival prediction.
* The use of low-resolution input rather than full patch processing leads to greatly reduced memory usage and time taken for inference.
* Leveraging the use of text-based PLIP encodings allows for solid zero-shot predictions on unannotated stroma and tumor classes.
* Ablation studies show the effects of each component: text and vision prompts for PLIP, homogeneity of the constructed graph based on node type, use of semantic features and negative edges for nodes.

Weaknesses

* The construction relies heavily on PLIP. This makes it hard to ablate the effect of the dataset used in pretraining. Previous heterogeneous graph methods such as [1] do not make use of PLIP. An experiment where the PLIP embeddings are progressively corrupted with noise would clearly illustrate how important they are in the learning process. I appreciate that it is difficult to measure the effect and the improvement that using PLIP embeddings could have on similar methods such as [1], but without these experiments its hard to quantify the actual improvement.
* The technical contributions appear to be relatively incremental:
* The use of VLM embeddings for zero-shot class prediction is common, here's an example with PLIP specifically [2].
* Building a heterogeneous graph for representing TME components has also been done, although with different classes and formulations [1], as also stated in the paper.
* The edge creation based on feature similarity with KNN rather than spatial positioning is novel, but minor.
* The choice of 50x downsampling appears to be somewhat arbitrary. An ablation study showing how much gain there is (or is not) based on higher/lower downsampling ratios (within hardware limits) would offer further insights.

[1] Wu, Junxian, et al. "Leveraging tumor heterogeneity: Heterogeneous graph representation learning for cancer survival prediction in whole slide images." Advances in Neural Information Processing Systems 37 (2024): 64312-64337.

[2] Guo, Miaotian, et al. "Multiple prompt fusion for zero-shot lesion detection using vision-language models." International Conference on Medical Image Computing and Computer-Assisted Intervention. Cham: Springer Nature Switzerland, 2023.

---

> ### Author Rebuttal · Authors · 2025-07-31
>
> Thanks for your constructive comment. Here are the responses to Weaknesses (W).
>
> **W1. The construction relies heavily on PLIP.**
>
> We thank the reviewer for such great comment. In fact, while your mentioned study [1] dose not use the PLIP embeddings, it relies on another foundation model i.e., UNI for patch-level feature extraction. For fair comparison between our method and the study in [1] , we apply the same encoder of VLM to extract features (i.e., PLIP and UNI embeddings), and the final survival prediction results are listed below:
>
> | Method            | Pre-trained VLM | BRCA                  | UCEC                  | LUAD                  | BLCA                  | Overall   |
> | ----------------- | --------------- | --------------------- | --------------------- | --------------------- | --------------------- | --------- |
> | ProtoSurv [1]     | PLIP            | 0.620 $\pm$ 0.012     | 0.698 $\pm$ 0.125     | 0.629 $\pm$ 0.035     | 0.624 $\pm$ 0.040     | 0.643     |
> | **ZTSurv (ours)** | **PLIP**        | **0.642 $\pm$ 0.029** | **0.726 $\pm$ 0.113** | **0.637 $\pm$ 0.033** | **0.637 $\pm$ 0.042** | **0.661** |
> | ProtoSurv [1]     | UNI             | 0.625 $\pm$ 0.009     | 0.705 $\pm$ 0.131     | 0.638 $\pm$ 0.026     | 0.629 $\pm$ 0.043     | 0.649     |
> | **ZTSurv (ours)** | **UNI**         | **0.650 $\pm$ 0.030** | **0.737 $\pm$ 0.115** | **0.646 $\pm$ 0.037** | **0.645 $\pm$ 0.038** | **0.670** |
>
> As shown in the table above, for both PLIP and UNI encoders, our ZTSurv consistently outperforms [1] across all datasets (BRCA, UCEC, LUAD, BLCA, and Overall), which demonstrates the effectiveness of our approach for survival prediction of human cancers from WSIs.
>
> [1] Wu, Junxian, et al. "Leveraging tumor heterogeneity: Heterogeneous graph representation learning for cancer survival prediction in whole slide images." Advances in Neural Information Processing Systems 37 (2024): 64312-64337.
>
> **W2. The technical contributions appear to be relatively incremental:**
>
> We appreciate your feedback and would like to to clarify several key points regarding our contributions at first, and then will address your comment point by point. The main contribution of this study is that we propose a novel end-to-end framework ZTSurv for survival prediction of human cancers that can segment different TME components and capture their spatial interactions for clinical outcome prediction. To the best of our knowledge, this is the first unified learning framework that can simultaneously conduct zero-shot tissue segmentation and survival prediction, which allows us to achieve better segmentation accuracy with low annotation burden and improve survival prediction performance, as these two tasks are highly related and thus can mutually enhance each other when learned together. More importantly, we achieve this on 50x down-sampled WSIs, which can significantly reduce the computational cost and storage burden in comparison with the existing studies that are implemented on high resolution whole-slide pathological images (WSIs).
>
> - **The use of VLM embeddings for zero-shot class prediction is common.**
>
>   While it is true that the VLM embeddings for zero-shot prediction have been explored, they mainly focus on classification or object detection tasks [2] [3], and it is still challenging to transfer the zero-shot learning ability of VLM to the tissue segmentation tasks on histopathological images and fewer studies work on it. Accordingly, we present a novel prototype guided zero-shot tissue segmentation model (Section 3.1 ) by the aid of patch-level PLIP embeddings that can achieve satisfied tissue segmentation results.
>
> - **Building a heterogeneous graph for representing TME components has been done.**
>
>   We acknowledge that heterogeneous graphs have been used to represent TME components for survival prediction in previous works, such as [1]. However, these methods follow a two-stage learning process where the TME patches should be firstly picked out from the whole-slide pathological images for the heterogeneous graph construction via training a tissue segmentation or patch-level classification model. Then, they rely on the constructed heterogeneous graph to build the clinical outcome prediction model. By contrast, we design an one stage learning paradigm that can simultaneously conduct the heterogeneous graph construction and survival prediction tasks in an end-to-end manner. The experimental results shown in Table 1 of the original manuscript clearly demonstrate the advantage of our method in comparison with the existing studies that treat the tasks of heterogeneous graph building and survival prediction independently.
>
> - **The edge creation is novel, but minor.**
>
>   Although the idea of concatenating the semantic information to the node visual embedding for edge representation is simple, the experimental results shown in Table 4 of the appendix clear show its advantage. Such observations also inspire to design more complicated ways to combine the semantic prompts with the visual representation of nodes for further performance improvement. We will point it out as the future research direction in the conclusion section of the revised manuscript.
>
> [1] Wu, Junxian, et al. "Leveraging tumor heterogeneity: Heterogeneous graph representation learning for cancer survival prediction in whole slide images." Advances in Neural Information Processing Systems 37 (2024): 64312-64337.
>
> [2] Guo, Miaotian, et al. "Multiple prompt fusion for zero-shot lesion detection using vision-language models." International Conference on Medical Image Computing and Computer-Assisted Intervention. Cham: Springer Nature Switzerland, 2023.
>
> [3] Kim, Sanghyun, et al. "Locality-Aware Zero-Shot Human-Object Interaction Detection." Proceedings of the Computer Vision and Pattern Recognition Conference. 2025.
>
> **W3. The choice of 50x downsampling appears to be somewhat arbitrary.**
>
> Thanks for this great comment. We chose 50x downsampling because the utilized TIL maps are labeled under such resolution [1], and thus we are unable to test the performance of our method on the TIL maps with higher resolution. By contrast, according to your suggestion, we conduct additional experiments to examine the impact of lower-resolution inputs by further downsampling the TIL map to 100×. The results are shown below:
>
> | Downsampling | BRCA              | UCEC              | LUAD              | BLCA              | Overall |
> | ------------ | ----------------- | ----------------- | ----------------- | ----------------- | ------- |
> | 50x          | 0.642 $\pm$ 0.029 | 0.726 $\pm$ 0.113 | 0.637 $\pm$ 0.033 | 0.637 $\pm$ 0.042 | 0.661   |
> | 100x         | 0.602 $\pm$ 0.027 | 0.682 $\pm$ 0.030 | 0.601 $\pm$ 0.028 | 0.603 $\pm$ 0.037 | 0.622   |
>
> As can be observed from the table above, training on the lower resolution TIL map will lead to the decrease for survival prediction. Such experimental results suggest that excessive downsampling reduces the amount of useful information available for survival prediction.
>
> [1] Saltz, Joel, et al. "Spatial organization and molecular correlation of tumor-infiltrating lymphocytes using deep learning on pathology images." Cell Reports 23.1 (2018): 181-193.e7.
>
>
>
> Once again, we sincerely thank you for your thoughtful review and constructive feedback. If you have any further questions, we would be glad to respond. If you find our responses satisfactory, we would greatly appreciate your reconsideration of the score.

---

> ### Author Response · Authors · 2025-08-06
>
> We sincerely appreciate the time and effort you have devoted to reviewing our submission. We have carefully prepared our rebuttal, aiming to address the valuable points you raised.  Should you have an opportunity to review our response, we would be grateful for any additional thoughts or clarifications you may wish to share. Thank you once again for your thoughtful and constructive feedback.
>
> Best regards,
>
> The Authors

---

> ### Comment · Area_Chair_c7nJ · 2025-08-07
> **Reminder: reviewer–author discussion is approaching its end**
>
> Dear Reviewer UaL5,
>
> This is a friendly reminder that the reviewer–author discussion phase is approaching its end, and we notice you have not yet responded in the discussion thread. Please take a moment to engage with the authors' rebuttal by sharing your thoughts or follow-up questions. If your concerns have been addressed, kindly acknowledge that in the discussion thread. Before submitting the “Mandatory Acknowledgement,” please ensure that you have engaged in actual communication with the authors. Thank you again for your time and efforts.
>
> Regards,
>
> AC

---

> ### Comment · Reviewer_UaL5 · 2025-08-08
>
> Many thanks to the authors for their detailed and considerate rebuttal.
> * W1: I thank the authors for showing that the model remains competitive even under embeddings different than PLIP.
> * W2: In terms of technical contribution, I'd still like to maintain my original points. In my perspective, the three points that the authors have addressed still remain and the novelty is incremental compared to previous work.
> * W3: The model being end-to-end and not working patchwise is consistently put forward as an advantage of the approach. While I agree that this is nice and practical, I find this coincidental with the fact that the TIL annotations are 50x downsampled, i.e. it is a limitation caused by the dataset. If full WSI-resolution TIL annotations were available, this direct end-to-end approach would not be possible, and I doubt that working at 50x downsampled resolution would lead to better results than using the full resolution and beat full-size foundation models.
> (I still appreciate that the model ends up having better survival prediction)
>
> With these in mind, I find it difficult to push my score up from 4 to 5, and keep it as 4. I greatly appreciate the authors' rebuttal and offer my sincere thanks.

---

> > ### Author Response · Authors · 2025-08-08
> >
> > We thank the reviewer for the thoughtful comments and continued engagement. We provide our responses regarding points W2 and W3 below.
> >
> > As demonstrated in [1], the spatial patterns of TILs are closely associated with patients’ survival, highlighting the power of spatial distribution analysis in addressing clinically relevant questions, even without high-resolution morphological detail. Additionally, compared to exitsting high-resolution patch-wise approaches as well as foundation models, our method benefits from jointly learning segmentation and survival prediction, allowing these tasks to support each other rather than being treated independently. This joint learning enables a more accurate characterization of the tumor microenvironment, which is crucial for improving prognosis.
> >
> > Finally, we again appreciate the reviewer’s valuable feedback and careful consideration.
> >
> > [1] Saltz, Joel, et al. "Spatial organization and molecular correlation of tumor-infiltrating lymphocytes using deep learning on pathology images." Cell Reports 23.1 (2018): 181-193.e7.

---

### Note · Authors · 2025-08-12

We sincerely thank all reviewers and ACs for their time, effort, and constructive feedback throughout the review process. We are encouraged that reviewers Ectr, FFJs, and WTnk recognized the originality of our work, and that all reviewers agreed the paper is clearly written and well-organized.

In response to Reviewer UaL5’s concerns, we emphasize that our main contribution is a unified end-to-end framework for zero-shot tissue segmentation and survival prediction on low-resolution WSIs. To our knowledge, this is the first approach to integrate both tasks in a zero-shot setting, reducing annotation burden and improving prediction performance while substantially lowering computational and storage costs compared to high-resolution methods. Regarding the concern of resolution, prior work [1] has shown that spatial patterns of TILs strongly correlate with survival, highlighting the feasibility of our low-resolution design. Moreover, unlike patch-wise methods and foundation models, our joint learning allows segmentation and survival prediction to reinforce each other, enabling a more accurate characterization of the TME for prognosis.

Regarding Reviewer Ectr’s points on comparison with slide-level foundation models, the impact of resolution, and the training strategies of baseline models, we have provided detailed clarifications and additional experiments in our rebuttal. Specifically, we demonstrated the feasibility and effectiveness of our low-resolution approach supported by spatial TIL patterns relevant to survival, and explained how our joint learning framework differs from patch-wise and foundation model pipelines.

We also responded in detail to Reviewer FFJs’s comments in rebuttal, covering aspects such as annotation burden, ground truth for unknown classes, the necrosis-inclusive ablation study, graph construction, LLM-generated descriptions, coefficients in the loss function, and verification that the survival loss backpropagates to segmentation.

Reviewer WTnk noted that most concerns have been addressed, including definition of zero-shot learning, verification of whether ZTSurv actually learns meaningful patterns, and the impact of low-quality or noisy initial TIL maps on performance.

Once again, we sincerely thank all reviewers and ACs for their constructive feedback and efforts in evaluating our work.

[1] Saltz et al. "Spatial organization and molecular correlation of tumor-infiltrating lymphocytes using deep learning on pathology images."

---

### Decision · Program_Chairs · 2025-09-17

**Decision:**

Accept (poster)

**Comment:**

This paper presents a method for cancer survival analysis that leverages zero-shot TME segmentation on low-resolution WSIs. The work is motivated by the need to reduce computational costs while maintaining prediction performance. Reviewers find the problem significant and the proposed framework creative, with the approach showing promising results for both segmentation and downstream survival prediction. While there are some concerns regarding the extent of experimental validation and the limited technical novelty, the rebuttal provides clarifications and additional results that address part of these issues. Considering that the paper tackles an important problem in computational pathology and makes a valuable contribution, it is recommended for acceptance, with the suggestion that the authors incorporate the reviewers’ feedback to further strengthen the final version.